# Spin-*s* *Q*-systems: Twist and open boundaries

**Yi-Jun He[1⋆], Jue Hou[1,2†§], Yi-Chao Liu[1‡] and Zi-Xi Tan[1∘]**

**1** School of physics & Shing-Tung Yau Center,
Southeast University, Nanjing 211189, P. R. China
**2** Mathematics Department, King's College London,
The Strand, London WC2R 2LS, United Kingdom

⋆ yjhe96@seu.edu.cn , † juehou@seu.edu.cn , ‡ iqna@seu.edu.cn , ∘ zxtan@seu.edu.cn

## Abstract

In integrable spin chains, the spectral problem can be solved by the method of Bethe Ansatz, which transforms the problem of diagonalization of the Hamiltonian into the problem of solving a set of algebraic equations named Bethe equations. In this work, we systematically investigate the spin-*s* XXX chain with twisted and open boundary conditions using the rational *Q*-system, which is a powerful tool to solve Bethe equations. We establish basic frameworks of the rational *Q*-system and confirm its completeness numerically in both cases. For twisted boundaries, we investigate the polynomiality conditions of the rational *Q*-system and derive physical conditions for singular solutions of Bethe equations. For open boundaries, we uncover novel phenomena such as hidden symmetries and boundary induced strings under specific boundary parameters. Hidden symmetries lead to the appearance of extra degeneracies in the Hilbert space, while the boundary induced string is a novel type of exact string configuration, whose length depends on the boundary magnetic fields. These findings, supported by both analytical and numerical evidences, offer new insights into the interplay between symmetries and boundary conditions.

| | |
|---|---|
| Received | 2025-03-04 |
| Accepted | 2025-07-30 |
| Published | 2025-08-27 |

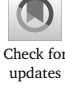

## Contents

---

§ Corresponding author.

# 1   Introduction

Quantum integrable models are distinguished by their capacity to yield exact solutions for a variety of physical quantities. Such solvability stems from the presence of a sufficient number of conserved quantities within the system. The Bethe Ansatz serves as a powerful tool to solve quantum integrable models, allowing physical quantities to be expressed in terms of solutions of the Bethe equations, which are a set of algebraic equations. However, the problem of whether the Bethe Ansatz produces all eigenstates of the Hamiltonian is by no means a simple task. To understand this question, a systematic analysis of Bethe roots and a method to generate all possible physical solutions are required.

A prototype of quantum integrable model is the spin-$\frac{1}{2}$ Heisenberg spin chain with the periodic boundary condition. It is important not only because of its potential to describe phenomena in the real world, but also its relationship with conformal field theories [1] and supersymmetric gauge theories like the $\mathcal{N} = 4$ super-Yang-Mills theory [2]. Notably, in the planar limit of $\mathcal{N} = 4$ super-Yang-Mills theory, the one-loop anomalous dimensions of certain single-trace operators map directly to the spectrum of this spin chain, such as the results shown in [2]. This relationship highlights the role of spin chain as a cornerstone for investigating the spectrum of operators in the $\mathcal{N} = 4$ super-Yang-Mills theory.

The study of the Bethe equations has a long history [3–16], dating back to the original work of Hans Bethe [17]. For the eigenstate with $M$ down-spins and $L-M$ up-spins, the corresponding Bethe equations are of the form

$$\left(\frac{u_k + \frac{i}{2}}{u_k - \frac{i}{2}}\right)^L = -\prod_{j=1}^M \frac{u_k - u_j + i}{u_k - u_j - i}, \qquad k = 1, \dots, M. \tag{1}$$

Typically, for given $L$ and $M$, this equation will have both physical solutions and non-physical solutions. 'Non-physical' here means that although they are solutions of the Bethe equation, the corresponding state constructed by Bethe Ansatz is not the eigenstate of the Hamiltonian. There are two kinds of non-physical solutions [12, 18]. One kind refers to the case when a Bethe solution characterizing a given state (we call it a solution) includes roots $\pm i/2$, which are called singular roots [6, 19]. The sequence of Bethe roots $\{+i/2, -i/2\}$ constitutes a singular string, which is a configuration that either appears in its entirety within a Bethe solution or is entirely absent.[1] The other kind refers to the case when any two roots in a Bethe solution coincide with each other, which are called repeated roots. To exclude these non-physical roots from solutions of the Bethe equation, additional conditions should be satisfied, which can be derived using the algebraic Bethe Ansatz [20]. These conditions are called physical conditions.

To exclude these non-physical solutions, various reformulations of the Bethe equation have been proposed, from which physical conditions can be easily derived. One of them is called the $TQ$-equation [4, 21–24]. By solving this equation, non-physical repeated roots are automatically eliminated. However, we still have non-physical singular roots. The physical conditions for these kinds of solutions are related to the polynomiality of the second solution of the $TQ$-relation, called the $P$-function. But to impose the polynomiality condition for the $P$-function in practice is not efficient for numerical calculations. Fortunately, there is another reformulation called the rational $Q$-system which can eliminate all the non-physical roots [25–33]. At the same time, the polynomiality conditions for those $Q$-functions appearing in the rational $Q$-system is efficient for numerical calculation. The numerical method based on this reformulation is much faster than directly solving the Bethe equations and turns out to be quite useful for obtaining all solutions of the Bethe equations at once for given $L$ and $M$.

The success of the rational $Q$-system in spin-$\frac{1}{2}$ XXX chain motivates the research of more general integrable spin chains [25–40]. In [33], the authors consider the spin-$s$ XXX chain with the periodic boundary condition, and they construct the generalized version of the rational $Q$-system and derive the corresponding physical conditions. Also in [27], the author generalize the construction of the rational $Q$-system to the spin-$\frac{1}{2}$ XXX chain with open boundary conditions. For spin-$s$ XXX chain with open boundary conditions, the spectrum has been discussed in [38].

The goal of this paper is to extend the construction of the rational $Q$-system to two important cases: the spin-$s$ XXX chain with twisted boundary conditions and the spin-$s$ XXX chain with open boundary conditions. These spin chains are related to the operator spectrum of $\gamma$-deformed $\mathcal{N} = 4$ super-Yang-Mills theory [41–43] and the correlators are related to open string states in the $\mathcal{N} = 4$ super-Yang-Mills theory [44–51], respectively. For both cases, we present the construction of the rational $Q$-systems. For the spin-$s$ XXX chain with twisted boundary conditions, we demonstrate the equivalence between the polynomiality of the $P$-function, introduced within the $\mathcal{T}Q$-system,[2] and the polynomiality of the rational $Q$-system. Through numerical evidence, we confirm the absence of physical singular solutions and physical solutions with repeated roots in this case, and check the completeness of the rational $Q$-system.

---

[1]This "all-or-none" behavior generalizes to higher spins: For a spin-$s$ system, the singular string is defined as the sequence $\{is, i(s-1), \dots, -is\}$, which similarly manifests as an indivisible unit in solutions.

[2]We use the notaion '$\mathcal{T}Q$' here instead of '$TQ$' because $\mathcal{T}$ refers to the $\mathcal{T}$-series introduced later in this paper.

For the spin-*s* XXX chain with open boundary conditions, we uncover intriguing phenomena such as hidden symmetries and the emergence of boundary induced string when specific constraints on the boundary parameters $\alpha, \beta$, spin-*s*, the length of chains $L$ and the magnon number $M$ are satisfied. Hidden symmetries arise when $\alpha - \beta$ is an integer, leading to extra degeneracies in the spectrum. These extra degeneracies is caused by operators denoted as $\hat{\mathfrak{q}}_\alpha^{p_0}$, which commute with the Hamiltonian in the Hilbert subspace with fixed magnon number $M$. The boundary induced strings can also arise when $\alpha - \beta := k \in \mathbb{Z}^+$, which is a novel type of exact string configuration related to boundary magnetic fields. They corresponds to the following set of roots:

$$\left\{ i\left(\alpha - \tfrac{1}{2}\right), i\left(\alpha - \tfrac{3}{2}\right), \ldots, i\left(\alpha - \tfrac{2k-1}{2}\right) \right\},$$
$$\text{and } \left\{ -i\left(\alpha - \tfrac{1}{2}\right), -i\left(\alpha - \tfrac{3}{2}\right), \ldots, -i\left(\alpha - \tfrac{2k-1}{2}\right) \right\}. \tag{2}$$

These are different from the usual string configuration in the case of the periodic boundary condition. These findings are supported by both analytical derivations and numerical results. The presence of hidden symmetries and boundary induced strings highlights the rich structure of integrable spin chains with open boundaries, offering new insights into their symmetries and Hilbert space.

The structure of the paper is as follows. In section 2, we review previous construction of the *Q*-system for the spin-*s* XXX chain with the periodic boundary condition [33]. In section 3, we present the spin-*s* *Q*-system for the spin-*s* XXX chain with twisted boundary conditions and check the completeness of the rational *Q*-system numerically. In section 4, we present the spin-*s* *Q*-system for the spin-*s* XXX chain with open boundary condition and also discuss the hidden symmetries and equators in detail both analytically and numerically. In section 5, we conclude this article and discuss future directions.

## 2 The periodic boundary condition

In this section, we review the framework of the rational *Q*-system for the spin-*s* XXX chain with the periodic boundary condition [33]. While this construction provides the starting point for our analysis, its generalization to twisted and open boundaries in subsequent sections will reveal novel physical phenomena, including hidden symmetries and emergent boundary induced strings.

### 2.1 The spin-*s* Bethe equation

The Hamiltonian of Heisenberg spin-$\frac{1}{2}$ XXX chain with the periodic boundary conditions $\mathbf{S}_{n+L} = \mathbf{S}_n$ is [52]

$$H = -\sum_{n=1}^{L} \left(\mathbf{S}_n \cdot \mathbf{S}_{n+1}\right), \tag{3}$$

for higher spin case, there will be some other terms, some explicit forms of Hamiltonian are listed in [33].

For the spin-*s* XXX chain, the corresponding Bethe equations have the following form

$$\left(\frac{u_k + is}{u_k - is}\right)^L = \prod_{j \neq k}^{M} \frac{u_k - u_j + i}{u_k - u_j - i}, \qquad k = 1, \ldots, M, \tag{4}$$

where $L$ is the length of the spin chain and $M$ is the number of magnons. Roughly speaking, there are two cases where non-physical solutions exist, as we mentioned in the introduction:

Table 1: $s = 1$.

| L | M | $\mathcal{N}_{BAE}$ | $\mathcal{N}_s(L,M)$ |
|---|---|---|---|
| 2 | 1 | 1 | 1 |
| 2 | 2 | 3 | 1 |
| 3 | 1 | 2 | 2 |
| 3 | 2 | 6 | 3 |
| 3 | 3 | 15 | 1 |
| 4 | 1 | 3 | 3 |
| 4 | 2 | 10 | 6 |
| 4 | 3 | 29 | 6 |

Table 2: $s = 3/2$.

| L | M | $\mathcal{N}_{BAE}$ | $\mathcal{N}_s(L,M)$ |
|---|---|---|---|
| 2 | 1 | 1 | 1 |
| 2 | 2 | 3 | 1 |
| 2 | 3 | 8 | 1 |
| 3 | 1 | 2 | 2 |
| 3 | 2 | 5 | 3 |
| 3 | 3 | 18 | 4 |

The first case is called singular solutions. In this case, the Bethe solution contains a string of Bethe roots in the form $\{is, i(s-1), \ldots, -is\}$. Since the energy of the Bethe state corresponding to a particular Bethe solution $\{u_1, \ldots, u_M\}$ is given by[3]

$$E \sim \sum_{j=1}^{M} \frac{s}{u_j^2 + s^2}, \tag{5}$$

a direct substitution of the Bethe solution into this expression yields divergent results, and the corresponding Bethe state is null. If some extra constraints, dubbed as physical constraints, are satisfied by the corresponding Bethe solutions, these null Bethe states can be regularized to yield eigenstates of the Hamiltonian. If physical constraints are not satisfied, the corresponding Bethe solution is known to be non-physical.

The second case is called repeated solutions. In this case, some of the Bethe roots in a Bethe solution are the same.[4] Applying the transfer matrix to the Bethe state corresponding to such kind of Bethe solution will lead to extra terms. The requirement that these extra terms should vanish also leads to extra constraints. Numerical evidence shows that after considering all possible physical conditions, the Bethe Ansatz can generate correct number of physical states. For the spin-$s$ XXX chain with the periodic boundary condition, the expected number of physical solutions for the sector with $M$ magnons is

$$\mathcal{N}_s(L,M) = c_s(L,M) - c_s(L,M-1), \tag{6}$$

where

$$c_s(L;M) = \sum_{j=0}^{\lfloor \frac{L+M-1}{2s+1} \rfloor} (-1)^j \binom{L}{j} \binom{L+M-1-(2s+1)j}{L-1}. \tag{7}$$

Note that the term $-c_s(L, M-1)$ in this counting equation is related to the number of $SU(2)$ descendant states.

We can check that the number of the solutions of Bethe Ansatz equations, $\mathcal{N}_{BAE}$, and the expected number of the solutions, $\mathcal{N}_s(L,M)$, satisfy the relation $\mathcal{N}_{BAE} \geq \mathcal{N}_s(L,M)$. And the solutions of Bethe Ansatz equations include repeated, singular, and regular solutions. In Table 1 and 2, we give examples to show that typically Bethe equations give more solutions than physical ones. These examples illustrate that Bethe equations typically yield too many solutions, which means that we need more constraints to eliminate all non-physical solutions.

---

[3] The tilde here means the equality up to a possible shift of constant.
[4] Note that a Bethe solution can be both singular and repeated.

## 2.2 The $TQ$-relation

The Bethe equations can be reformulated into the following $TQ$-relation

$$\tau(u)Q(u) = (u+is)^L Q(u-i) + (u-is)^L Q(u+i),\qquad(8)$$

where $\tau(u)$ is the eigenvalue of the transfer matrix. This is a polynomial of degree $L$. For a Bethe solution $\{u_1, u_2, \ldots, u_M\}$, the $Q$-function is defined to be

$$Q(u) = \prod_{j=1}^{M}(u-u_j).\qquad(9)$$

By taking $u = u_k$ for $k = 1, \ldots, M$, the $TQ$-relation becomes that

$$\frac{(u_k+is)^L Q(u_k-i)}{(u_k-is)^L Q(u_k+i)} = -1,\qquad(10)$$

which are Bethe equations. For the case of repeated roots, there will be a higher order zero on the left hand side of the $TQ$-relation. Thus there should be a zero of the same order on the right hand side of the $TQ$-relation, which leads to more constraints. These are just the physical conditions for the repeated roots.

Since the $TQ$-relation (8) is a second-order difference equation for the function $Q(u)$, there should be another solution which is linearly independent of the function $Q(u)$. We call it the $P$-function $P(u)$. One can prove that the $Q$-function and the $P$-function satisfy the equation

$$P\left(u+\tfrac{i}{2}\right)Q\left(u-\tfrac{i}{2}\right) - P\left(u-\tfrac{i}{2}\right)Q\left(u+\tfrac{i}{2}\right) = \prod_{k=0}^{2s-1}\left(u-i\left(s-k-\tfrac{1}{2}\right)\right)^L,\qquad(11)$$

which is called the Wronskian relation. Generally speaking, the $P$-function is not a polynomial and also not unique. However, one can proof that the additional requirement of $P(u)$ being a polynomial implies the physical conditions for the singular roots [33]. Thus, the $TQ$-relation with the requirement that $\tau(u)$, $Q(u)$ and $P(u)$ are polynomials will give all constraints to select all the physical solutions.

## 2.3 The rational $Q$-system

Although it is possible to impose all needed extra constraints in the formulation of the $TQ$-relation, the situation is not so satisfactory since numerically it is not efficient to impose the polynomiality of the function $P(u)$. Fortunately, there is another way to reformulate it, called the rational $Q$-system. In this part, we will review the construction of the rational $Q$-system for the spin-$s$ XXX chain.

The the rational $Q$-system is a system of $Q$-functions defined on nodes of a Young diagram with $M$ boxes in the first row and $2sL - M$ boxes in the second row, as shown in Figure 1. On each node of the Young diagram there is a corresponding $Q$-function denoted as $Q_{a,b}$.

Such a system of $Q$-functions satisfies the following three requirements:

**The $QQ$-relation.** The $Q$-functions satisfy the $QQ$-relations

$$Q_{a,b+1}Q_{a+1,b} = Q_{a,b}^+ Q_{a+1,b+1}^- - Q_{a,b}^- Q_{a+1,b+1}^+.\qquad(12)$$

Here the notation $f^{\pm}(u)$ is defined to be $f^{\pm}(u) \equiv f(u \pm \tfrac{i}{2})$. One can see that for each box of the Young diagram there is an associated $QQ$-relation.

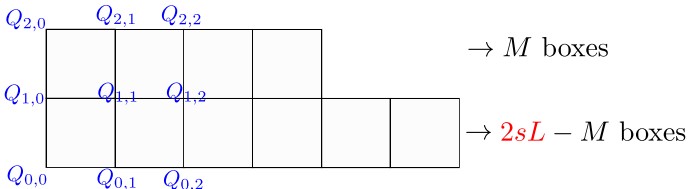

Figure 1: The Young diagram corresponding to the rational $Q$-system. There is an $Q$-function on vertices of each square and an $QQ$-relation associated to each box. The subscript of the notation $Q_{a,b}$ encodes the location: $a, b$ denotes the $a$-th row from the bottom to the top and the $b$-th column from the left to the right.

**Boundary conditions.** Introducing the functions[5]

$$\gamma(u) \equiv \prod_{k=1}^{2s-1}(u - i(s-k))^L, \qquad q_\gamma(u) \equiv \prod_{k=0}^{2s-1}(u - i(s-k-\tfrac{1}{2}))^L, \tag{13}$$

then boundary conditions of the $Q$-system fix the $Q$-functions at upper and left boundaries of the Young diagram in the following way

$$\begin{aligned} Q_{0,0}(u) &= q_\gamma(u), & Q_{1,0}(u) &= Q(u), \\ Q_{2,b}(u) &= 1, & b &= 0, 1, \ldots, M. \end{aligned} \tag{14}$$

**Analyticity requirements.** $Q_{0,1}/\gamma$ and all other $Q$-functions should be polynomials. Actually, this is not the minimal analyticity requirement of the rational $Q$-system. It has been proved that once we require $Q_{0,1}/\gamma$ and $Q_{0,2s+1}$ to be polynomial, the full analyticity requirement will be automatically satisfied [33]. The minimal analyticity requirement is that $Q_{0,1}/\gamma$ and $Q_{0,2s+1}$ are polynomials.

Here we shortly explain why the rational $Q$-system works. Taking the two $QQ$-relations associated to the first column of the Young diagram, we have

$$\begin{aligned} Q_{1,1} &= Q_{1,0}^+ - Q_{1,0}^-, \\ Q_{1,0}Q_{0,1} &= Q_{0,0}^+ Q_{1,1}^- - Q_{0,0}^- Q_{1,1}^+. \end{aligned} \tag{15}$$

Taking $u = u_k$ for $k = 1, \ldots, M$ and using the fact that $Q_{1,0}(u_k) = 0$, it is easy to get Bethe equations from these $QQ$-relations. Also, it has been proved that requiring $Q_{0,1}/\gamma$ and $Q_{0,2s+1}$ to be polynomial implies that the second solution of the $TQ$-relation $P(u)$ is also a polynomial, thus all required constraints for physical solutions are also contained in the rational $Q$-system.

To solve the rational $Q$-system numerically, one could follow the steps below:

1. Make a polynomial Ansatz for the function $Q_{1,0}$:

$$Q_{1,0}(u) = u^M + \sum_{k=0}^{M-1} c_k u^k, \tag{16}$$

   where the coefficients $c_k$ are to be solved numerically.

2. Solve the $Q$-functions $Q_{1,n}$ iteratively using the $QQ$-relation

$$Q_{1,n}(u) = Q_{1,n-1}(u + \tfrac{i}{2}) - Q_{1,n-1}(u - \tfrac{i}{2}). \tag{17}$$

---

[5]We define $\gamma(u) = 1$ when $s = \tfrac{1}{2}$.

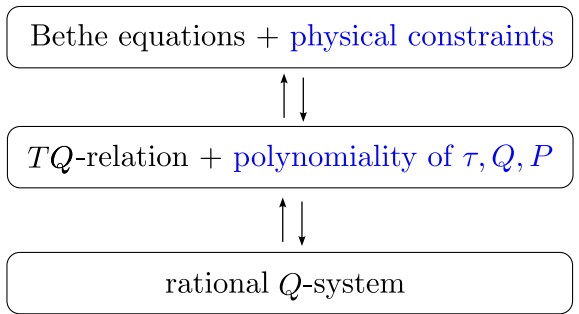

Figure 2: Relations among Bethe equations, the $TQ$-relation, and the rational $Q$-system.

3. Solve the $Q$-functions $Q_{0,n}$ iteratively using the $QQ$-relation

$$Q_{0,n}(u) = \frac{Q_{0,n-1}(u+\frac{i}{2})Q_{1,n}(u-\frac{i}{2}) - Q_{0,n-1}(u-\frac{i}{2})Q_{1,n}(u+\frac{i}{2})}{Q_{1,n-1}(u)}. \tag{18}$$

4. Solve the zero reminder equations[6] for $Q_{0,n}$ where $n = 1, \ldots, 2s+1$ when $M \geq 2s+1$, otherwise solve zero reminder equation for $Q_{0,1}$. Numerical values of $c_k$ ($k = 1 \ldots, M-1$) will be obtained.

5. Solve the equation $Q_{1,0}(u) = 0$ to obtain the Bethe roots.

The relations between Bethe equations, the $TQ$-relation, and the rational $Q$-system can be summarized in Figure 2.

## 2.4 From the $Q$-system to $\mathcal{T}Q$-relations

For later convenience, we introduce the $\mathcal{T}$-series and a sequence of $\mathcal{T}Q$-relations.[7] Define

$$\mathcal{T}_n = Q_{0,n}^+ + Q_{0,n}^- - Q_{0,n+1}, \qquad n = 0, 1, \ldots, \tag{19}$$

and from $QQ$-relations we know that

$$\mathcal{T}_n Q_{1,n} = Q_{0,n}^+ Q_{1,n}^{--} + Q_{0,n}^- Q_{1,n}^{++}, \qquad n = 0, \ldots, 2sL - M. \tag{20}$$

It is proven that $\mathcal{T}_0$ is related to the eigenvalue of the transfer matrix $\tau$ in the following way [33]

$$\mathcal{T}_0(u) = \tau(u)\gamma(u). \tag{21}$$

## 3 Twisted boundary conditions

In this section we present the construction of the rational $Q$-system for closed spin-$s$ XXX chain with twisted boundary condition. We also prove the equivalence between the polynomiality of $P$-function and the polynomiality of $Q$-functions $Q_{0,1}/\gamma$, $Q_{0,2s+1}$. Numerically we find that the rational $Q$-system gives the correct number of eigenstates for spin-$s$ XXX chain with twisted boundary condition.

---

[6]In order to keep the polynomiality of $Q_{0,n}(u)$, the remainder on the right hand side should be zero, which results in what we call the zero remainder equation.

[7]We use the notation $\mathcal{T}_n$ here to distinguish it from the function $\tau$ appearing in the previously introduced $TQ$-relation. They are related by equation (21).

Table 3: For $L = 5, M = 2, s = 1/2, \theta = 1/3$, there are 5 repeated solutions, 1 singular solution and 10 regular solutions from Bethe Ansatz equations.

| repeated | {1.733,1.733},{-1.38,-1.38},{0.389,0.389},{-0.338,-0.338},{0.017,0.017} |
|---|---|
| singular | $\{-1/2i, 1/2i\}$ |
| regular | $\{-11.989 - 6.007i, -11.989 + 6.007i\}, \{-9.117, 0.169\},$ $\{-8.903, -0.157\}, \{-9.423, 0.708\}, \{-8.499, -0.677\},$ $\{0.385 - 0.506i, 0.385 + 0.506i\}, \{-0.03, 0.47\},$ $\{-0.249 - 0.501i, -0.249 + 0.501i\}, \{-0.406, 0.061\}, \{-0.468, 0.535\}$ |

### 3.1 Twisted boundary conditions

For a length-$L$ spin chain, twisted boundary conditions are given by

$$S_{L+1}^z = S_1^z, \quad S_{L+1}^\pm = e^{\pm i\theta} S_1^\pm, \tag{22}$$

where $S_n^{\pm,z}$ are generators of $\mathfrak{sl}(2)$ acting on site-$n$ and with the same Hamiltonian (3) when $s = \frac{1}{2}$. There is also another equivalent form of Hamiltonian($s = \frac{1}{2}$) [53]

$$H_\theta = 4 \sum_{n=1}^{L} \left( \frac{1}{4} - S_n^z S_{n+1}^z - \frac{e^{i\theta/L}}{2} S_n^+ S_{n+1}^- - \frac{e^{i\theta/L}}{2} S_n^- S_{n+1}^+ \right), \tag{23}$$

with $S_n^\pm = S_n^x \pm i S_n^y$ and the periodic boundary condition $S_{L+1}^{\pm,z} = S_1^{\pm,z}$.

The twisted boundary condition will break the $SU(2)$ symmetry of the Hamiltonian with the periodic boundary condition to the $U(1)$ symmetry. XXX chain with twisted boundary conditions can be solved by Bethe Ansatz, and the corresponding Bethe equations are

$$e^{i\theta} \left( \frac{u_k + is}{u_k - is} \right)^L \prod_{j=1, j \neq i}^{M} \frac{u_k - u_j - i}{u_k - u_j + i} = 1, \qquad k = 1, \dots, M. \tag{24}$$

However, not all solutions from Bethe equations are physical. The solutions from Bethe equations include repeated solutions, singular solutions and regular solutions. The type of solutions for twisted boundary conditions is same as the type for periodic boundary conditions. Table 3 presents an example of Bethe solutions. For $L = 5, M = 2, s = 1/2, \theta = 1/3$, there are 16 solutions from Bethe Ansatz equations. However, according to the completeness, for $U(1)$ symmetric case, the expected number of solutions should be

$$\mathcal{N}_s^{U(1)}(L, M) = c_s(L, M), \tag{25}$$

where $c_s(L; M)$ is given by (7). For the case in Table 3, the expected number of solutions is 10. The Bethe Ansatz equations give too many solutions. So we need the $Q$-system to eliminate all non-physical solutions.[8]

---

[8]The Bethe states about non-physical solutions are not eigenstates of the Hamiltonian.

## 3.2 The rational $Q$-system

The $QQ$-relation for the twisted version of the rational $Q$-system is given by

$$Q_{a+1,b}Q_{a,b+1} = \kappa_a Q_{a,b}^+ Q_{a+1,b+1}^- - Q_{a,b}^- Q_{a+1,b+1}^+, \tag{26}$$

where $\kappa_0 = \kappa \equiv e^{i\theta}$, $\kappa_1 = 1$. The boundary conditions here are the same as the twistless case reviewed in Section 2:[9]

$$Q_{0,0}(u) = q_\gamma(u), \qquad Q_{1,0}(u) = Q(u) = \prod_{j=1}^{M}(u - u_j) = u^M + \sum_{k=0}^{M-1} c_k u^k,$$
$$Q_{2,b}(u) = 1, \qquad b = 0, 1, \dots, M. \tag{27}$$

We expect that the minimal analyticity requirement should also be the same, *i.e.*, $Q_{0,1}/\gamma$ and $Q_{0,2s+1}$ are polynomials.

The twisted version of $QQ$-relations also encode twisted Bethe equations. Taking the $QQ$-relations associated to the first column of the Young diagram, we have

$$Q_{1,1} = Q_{1,0}^+ - Q_{1,0}^-,$$
$$Q_{1,0}Q_{0,1} = \kappa Q_{0,0}^+ Q_{1,1}^- - Q_{0,0}^- Q_{1,1}^+. \tag{28}$$

Taking $u = u_k$ for $k = 1, \dots, M$, we have

$$\kappa \frac{Q_{0,0}^+(u_k)Q_{1,0}^{--}(u_k)}{Q_{0,0}^-(u_k)Q_{1,0}^{++}(u_k)} = -1. \tag{29}$$

This is just the twisted Bethe equation (24).

## 3.3 From the $Q$-system to $\mathcal{T}Q$-relations

Define the $\mathcal{T}$-series in the following way

$$\mathcal{T}_n \equiv \kappa Q_{0,n}^+ + Q_{0,n}^- - Q_{0,n+1}, \tag{30}$$

then by $QQ$-relations we have

$$\mathcal{T}_n Q_{1,n} = \kappa Q_{0,n}^+ Q_{1,n}^{--} + Q_{0,n}^- Q_{1,n}^{++}. \tag{31}$$

Taking $n = 0$ we have

$$\mathcal{T}_0 Q_{1,0} = \kappa Q_{0,0}^+ Q_{1,0}^{--} + Q_{0,0}^- Q_{1,0}^{++}. \tag{32}$$

Using the fact that

$$Q_{0,0}^-(u) = q_\gamma^-(u) = \gamma(u)(u - is)^L, \qquad Q_{0,0}^+(u) = q_\gamma^+(u) = \gamma(u)(u + is)^L, \tag{33}$$

we have

$$\frac{\mathcal{T}_0}{\gamma} Q_{1,0} = \kappa(u + is)^L Q_{1,0}^{--} + (u - is)^L Q_{1,0}^{++}. \tag{34}$$

If $\mathcal{T}_0/\gamma$ is a polynomial, then by taking $u = u_k$ we obtain the twisted Bethe equations.

---

[9] The value of $\theta$ is taken in the sense of modulo $2\pi$.

## 3.4 The P-function

In this part we introduce the $P$-function in the twisted case and give the explicit expression for it.

In the case of the periodic boundary condition, the $P$-function is defined to be the second solution of the $\mathcal{T}Q$-relation. Here the $P$-function is defined to be the solution of the following equation

$$\mathcal{T}_0 P = Q_{0,0}^+ P^{--} + \kappa Q_{0,0}^- P^{++}, \tag{35}$$

which is slightly different from (32). Together with (32) we know that the $P$-function and $Q$-function satisfy the Wronskian relation

$$q_\gamma = \begin{vmatrix} Q^- & Q^+ \\ \kappa^{-1} P^- & P^+ \end{vmatrix}. \tag{36}$$

We can express $Q_{0,1}$ and $\mathcal{T}_0$ by $Q$ and $P$ by using $QQ$-relations (26) and $\mathcal{T}Q$-relations (31)

$$Q_{0,1} = \begin{vmatrix} (DQ)^- & (DQ)^+ \\ (D_\kappa P)_\kappa^- & (D_\kappa P)_\kappa^+ \end{vmatrix},$$
$$\mathcal{T}_0 = \begin{vmatrix} Q^{--} & Q^{++} \\ P_\kappa^{--} & P_\kappa^{++} \end{vmatrix}, \tag{37}$$

where we introduced the notations

$$f_\kappa^{[+n]} \equiv \kappa^{\frac{n}{2}} f^{[+n]},$$
$$f^{[\pm n]}(u) \equiv f(u \pm \tfrac{in}{2}), \tag{38}$$
$$D_\kappa f \equiv f_\kappa^+ - f_\kappa^- = \kappa^{\frac{1}{2}} f^+ - \kappa^{-\frac{1}{2}} f^-.$$

When $\kappa = 1$, we have $D_\kappa f = Df = f^+ - f^-$. It is easy to check the following identity:

$$(D_{\kappa_1} f)_{\kappa_2}^+ = D_{\kappa_1}(f_{\kappa_2}^+), \qquad \forall \kappa_1, \kappa_2. \tag{39}$$

Then, using the mathematical induction and $\mathcal{T}Q$-relations (31), we can express all the $Q$-functions and $P$-functions in terms of $Q$ and $P$:

$$Q_{1,n} = D^n Q,$$
$$Q_{0,n} = \kappa^{\frac{n-1}{2}} \begin{vmatrix} (D^n Q)^- & (D^n Q)^+ \\ (D_\kappa^n P)_\kappa^- & (D_\kappa^n P)_\kappa^+ \end{vmatrix},$$
$$\mathcal{T}_n = \kappa^{\frac{n}{2}} \begin{vmatrix} (D^n Q)^{--} & (D^n Q)^{++} \\ (D_\kappa^n P)_\kappa^{--} & (D_\kappa^n P)_\kappa^{++} \end{vmatrix}. \tag{40}$$

Now we construct the explicit expression for general $P$-function. The Wronskian relation (36) can be rewritten as

$$q_\gamma = Q_{0,0} = \kappa^{-\frac{1}{2}} \begin{vmatrix} Q^- & Q^+ \\ P_\kappa^- & P_\kappa^+ \end{vmatrix}. \tag{41}$$

Plugging the Wronskian relation into the $\mathcal{T}Q$-relation (31), we have

$$\begin{aligned}
\frac{\mathcal{T}_0}{Q^{++} Q^{--}} &= \frac{\kappa q_\gamma^+}{Q^{++} Q} + \frac{q_\gamma^-}{Q Q^{--}} \\
&= \left( \frac{\kappa P^{++}}{Q^{++}} - \frac{P}{Q} \right) + \left( \frac{P}{Q} - \frac{\kappa^{-1} P^{--}}{Q^{--}} \right) \\
&\equiv S_\kappa^+ + S_\kappa^-,
\end{aligned} \tag{42}$$

where we introduced the function $S$

$$S \equiv \frac{\kappa^{\frac{1}{2}}P^+}{Q^+} - \frac{\kappa^{-\frac{1}{2}}P^-}{Q^-}\,,\tag{43}$$

and the notation

$$f(Q,P)^{[+n]}_\kappa \equiv f(Q^{[+n]}, \kappa^{\frac{n}{2}}P^{[+n]})\,.\tag{44}$$

Using the Wronskian relation (36), we have

$$
\begin{aligned}
S^+_\kappa &\equiv \frac{\kappa P^{++}}{Q^{++}} - \frac{P}{Q} = \frac{\kappa q^+_\gamma}{Q^{++}Q}\,,\\
S^-_\kappa &\equiv \frac{P}{Q} - \frac{\kappa^{-1}P^{--}}{Q^{--}} = \frac{q^-_\gamma}{QQ^{--}}\,.
\end{aligned}\tag{45}
$$

Numerical evidence shows that for $\theta \neq 0$, if $\mathcal{T}_0/\gamma$ is a polynomial, then there is no repeated root given by zeros of the function $Q$.[10] Thus $Q$ can be written as

$$
\begin{aligned}
Q(u) &= Q_s(u)Q_r(u)\,,\\
Q_s(u) &= \prod_{j=0}^{2s}[u - i(s-j)]\,,
\end{aligned}\tag{46}
$$

where $Q_r$ does not have singular roots. The function $S^-_\kappa$ can be decomposed in the following way

$$S^-_\kappa = \frac{q^-_\gamma}{QQ^{--}} = \pi(u) + \frac{q_1(u)}{Q_r} + \frac{q_2(u)}{Q^{--}_r} + \frac{a_1}{u+is} + \frac{a_2}{u-i(s+1)}\,.\tag{47}$$

In the above expression, $\pi(u)$, $q_1(u)$ and $q_2(u)$ are polynomials and the degrees of $q_1(u)$, $q_2(u)$ are less than the degree of $Q_r$. $a_1, a_2$ are constants. For any given polynomial $\pi(u)$, there always exists a polynomial $\rho(u)$, such that we can decompose $\pi(u)$ as $\pi(u) = \rho(u) - \kappa^{-1}\rho(u-i)$. Plugging (47) into (42), we have

$$
\begin{aligned}
\frac{\mathcal{T}_0}{Q^{++}_r Q^{--}_r Q^{++}_s Q^{--}_s} =& \left( \pi + \frac{q_1}{Q_r} + \frac{q_2}{Q^{--}_r} + \frac{a_1}{u+is} + \frac{a_2}{u-i(s+1)} \right)\\
&+ \kappa\left( \pi^{++} + \frac{q^{++}_1}{Q^{++}_r} + \frac{q^{++}_2}{Q_r} + \frac{a_1}{u+i(s+1)} + \frac{a_2}{u-i(s)} \right)\,.
\end{aligned}\tag{48}
$$

Taking the residue at zeros of $Q_r$ on both sides of this equation, we have

$$q_1 + \kappa q^{++}_2 = 0\,,\qquad \text{at zeros of } Q_r\,.\tag{49}$$

Since degrees of polynomials $q_1$, $q_2$ are less than the degree of $Q_r$, it is not hard to see that the above equation actually holds at all $u \in \mathbb{C}$. Then we have

$$S^-_\kappa = \rho - \kappa^{-1}\rho^{--} + \frac{q_1}{Q_r} - \frac{\kappa^{-1}q^{--}_1}{Q^{--}_r} + \frac{a_1}{u+is} + \frac{a_2}{u-i(s+1)}\,.\tag{50}$$

---

[10]We have verified this for several specific cases, including $L = 6$, $M = 4$, $s = 1$ with various generic twists. These examples strongly support our conclusion. However, for larger systems, numerical confirmation becomes increasingly computationally intensive, limiting our ability to extend the verification to more extensive parameter sets.

For the last two terms on the right hand side, we have

$$
\begin{aligned}
&\frac{a_1}{u+is} + \frac{a_2}{u-i(s+1)} \\
&= \left( \frac{a_1}{u+is} - \frac{\kappa a_2}{u-is} \right) - \kappa^{-1} \left( \frac{a_1}{u+i(s-1)} - \frac{\kappa a_2}{u-i(s+1)} \right) + \frac{\kappa^{-1} a_1}{u+i(s-1)} + \frac{\kappa a_2}{u-is} \\
&= \left( \frac{a_1}{u+is} - \frac{\kappa a_2}{u-is} - i\kappa^{-1} a_1 \psi_\kappa(-iu+s) - i\kappa a_2 \psi_\kappa(-iu-s+1) \right) \\
&\quad - \kappa^{-1} \left( \frac{a_1}{u+i(s-1)} - \frac{\kappa a_2}{u-i(s+1)} - i\kappa^{-1} a_1 \psi_\kappa(-iu+s-1) - i\kappa a_2 \psi_\kappa(-iu-s) \right),
\end{aligned}
\tag{51}
$$

where the function $\psi_\kappa$ is defined as

$$
\psi_\kappa(z) \equiv -\gamma_E + \sum_{n=0}^{\infty} \kappa^{n+1} \left( \frac{1}{n+1} - \frac{1}{n+z} \right).
\tag{52}
$$

Note that $\gamma_E$ is the Euler constant. The function $\psi_\kappa$ satisfies the equation

$$
\psi_\kappa(z+1) - \kappa^{-1} \psi_\kappa(z) = \frac{1}{z}.
\tag{53}
$$

For $|\kappa| \le 1$, the summation in (52) is convergent.[11] Since we only consider the case when $\theta$ is real, this condition is satisfied.

Comparing (45) and the expansion (50), we know that the form of $P$-function should be like

$$
P = P_0 + Q \left( -i\kappa^{-1} a_1 \psi_\kappa(-iu+s) - i\kappa a_2 \psi_\kappa(-iu-s+1) \right),
\tag{54}
$$

where $P_0$ is a polynomial of $u$. Here we used the equation (51). The expression for $P$ (54) can be also written in the form

$$
P(u) = P_0(u) + Q(u) \sum_{j=-(s-1)}^{s} \tilde{a}_j \psi_\kappa(-iu+j).
\tag{55}
$$

For completeness, we emphasize that in case there are repeated roots when $\mathcal{T}_0/\gamma$ is a polynomial, the $P$-function can be written as[12]

$$
P(u) = P_0(u) + Q(u) \sum_{m=0}^{M-1} \sum_{j=-(s-1)}^{s} a_j^{(m)} \psi_\kappa^{(m)}(-iu+j),
\tag{56}
$$

which goes back to the expression for $P$ in the twistless case [33] when taking $\theta \to 0$. On the right hand side, $a_j^{(m)}$ are again constants and functions $\psi_\kappa^{(m)}(z)$ are defined as

$$
\psi_\kappa^{(m)}(z) \equiv \frac{d^m}{dz^m} \psi_\kappa(z) = (-1)^{m+1} m! \sum_{n=0}^{\infty} \kappa^{n+1} \frac{1}{(n+z)^{m+1}},
\tag{57}
$$

which satisfy the equation

$$
\psi_\kappa^{(m)}(z+1) - \psi_\kappa^{(m)}(z) = \frac{(-1)^{m+1} m!}{z^{m+1}} \psi_\kappa^{(m)}(z+1) - \psi_\kappa^{(m)}(z) = \frac{(-1)^{m+1} m!}{z^{m+1}}.
\tag{58}
$$

---

[11] A solution of the difference equation $F(z+1) - \kappa^{-1} F(z) = \frac{1}{z}$ is given by

$$
F(z) = -\kappa \Phi(\kappa, 1, z) + \kappa^{1-z} c_1(\kappa),
$$

where $\Phi(\kappa, a, z)$ is the Lerch transcendent and $c_1(\kappa)$ is an arbitrary function about $\kappa$ but independent of $z$. However, $\Phi(\kappa, 1, z)$ is convergent with $|\kappa| < 1$ but divergent when $\kappa = 1$. To taking the twistless limit, $\kappa \to 1$, we introduce the function $\psi_\kappa(z)$.

[12] This can be derived in the same way as [33].

**Polynomiality.** Now we prove that the following two statements are equivalent to each other:

(i) $Q_{0,1}/\gamma$ and $Q_{0,2s+1}$ are polynomials.

(ii) The function $P$ is a polynomial.[13]

From the definition of $\psi_\kappa$ (52), we know that the singular points of $\psi_\kappa$ are $z = 0, -1, -2, \ldots$ Therefore by (55), possible poles of $P$ are given by $u = i(s-1), i(s-2), i(s-3), \ldots$

By the $\mathcal{T}P$-relation (35), we have the relation

$$\frac{\mathcal{T}_0}{\gamma} P = (u + is)^L P^{--} + \kappa (u - is)^L P^{++}. \tag{59}$$

Using the condition that $\mathcal{T}_0/\gamma$ is a polynomial and taking $u = is$, we know from (59) that the function $P$ has no pole at $u = i(s-1)$. Then taking $u = i(s-1)$ for (59), we know that the function $P$ should be regular at $u = i(s-2)$. The same thing can be done till taking $u = -i(s-1)$, from which we know that $P$ have no pole at $u = -is$. If we try to take $u = -is$ for (59), such kind of arguments breaks down. However, just as the process given in [33], we can actually prove that if $\mathcal{T}_{2s}$ is a polynomial, then the function $P$ is regular at $u = -i(s+1)$. Starting with this result and using (59), it is easy to see that $P$ is regular at $u = -i(s+2), -i(s+3), \ldots$. Thus we conclude that if $\mathcal{T}_0/\gamma$ and $\mathcal{T}_{2s}$ are polynomials, then function $P$ is a polynomial.

By definition of $\mathcal{T}_n$ functions (30), it is easy to see that the condition $Q_{0,1}/\gamma$ is a polynomial is equivalent to the condition that $\mathcal{T}_0/\gamma$ is a polynomial. Actually, one can further prove that if $Q_{0,1}/\gamma$ is a polynomial then $\mathcal{T}_1, \ldots, \mathcal{T}_{2s-1}$ are also polynomials [33]. Then by (19), we know that $\mathcal{T}_{2s}$ is also a polynomial if we further assume that $Q_{0,2s+1}$ is a polynomial. Thus if $Q_{0,1}/\gamma$ and $Q_{0,2s+1}$ are polynomials, then function $P$ is a polynomial.

Assuming that function $P$ is a polynomial, then by (40), all $Q$-functions are polynomials. Then we conclude that the statement (i) and the statement (ii) are equivalent.

## 3.5 Physical conditions

Under twisted boundary conditions, we propose that if a solution containing repeated roots fails to yield a polynomial $Q_{0,1}/\gamma$ in the untwisted limit ($\theta = 0$), these repeated roots persist for any non-zero twist angle $\theta$. It can be demonstrated through a proof by contradiction. Assume, for contradiction, that these repeated roots aligned into distinct single roots when $\theta \neq 0$, then $Q_{0,1}/\gamma$ must remain a polynomial for all $\theta$ due to the $\mathcal{T}Q$-relation, including the limit $\theta \to 0$. This directly contradicts the initial assumption that $Q_{0,1}/\gamma$ is non-polynomial at $\theta = 0$. Consequently, the repeated roots cannot aligned and must persist for any $\theta \neq 0$.

In the following, we also assume that twisted boundary conditions is a strong enough regularization scheme such that there is no repeated root if $Q_{0,1}/\gamma$ is required to be a polynomial.

Now we derive physical conditions for Bethe solutions with singular roots. The Wronskian relation (36) can be rewritten as

$$\frac{P}{Q} - \frac{\kappa^{-1} P^{--}}{Q^{--}} = \frac{q_\gamma^-}{Q Q^{--}}, \tag{60}$$

from which we can derive that

$$\frac{P^{[-2s-2]}}{Q^{[-2s-2]}} = \frac{P^{[2s+2]}}{Q^{[2s+2]}} \kappa^{2s+2} - \sum_{k=0}^{2s+1} \frac{\kappa^{2s+2-k} q_\gamma^{[2s+1-2k]}}{Q^{[2s+2-2k]} Q^{[2s-2k]}}, \tag{61}$$

---

[13]Here $P$ satisfies $\mathcal{T}P$-relation (35), which includes the Bethe Ansatz equations.

which can be rewritten as

$$P^{[-2s-2]}Q^{[2s+2]} = P^{[2s+2]}Q^{[-2s-2]}\kappa^{2s+2} - Q^{[-2s-2]}Q^{[2s+2]}\mathbf{B}, \tag{62}$$

where

$$\mathbf{B} \equiv \sum_{k=0}^{2s+1} \frac{\kappa^{2s+2-k}q_\gamma^{[2s+1-2k]}}{Q^{[2s+2-2k]}Q^{[2s-2k]}}. \tag{63}$$

Taking $u = 0$, since $Q^{[2s+2]}(0) \neq 0$, we know that regularity of function $P(u)$ at $u = -i(s+1)$ is equivalent to the regularity of $\mathbf{B}(u)$ at $u = 0$. Since for twisted boundary conditions with general $\kappa$, numerical evidence shows that there is no repeated solution such that $\mathcal{T}_0/\gamma$ is a polynomial, we can write $Q$ as $Q = (u - is)(u - i(s - 1)) \cdots (u + is)Q_r$. Taking $u = \epsilon$ where $\epsilon \to 0$, we have

$$\mathbf{B}(\epsilon) = \frac{1}{\epsilon} \frac{(i)^{L-2}(2i)^{L-2}(3i)^{L-2} \cdots (2si)^{L-2}}{i(2s+1)} \left[ \frac{\kappa^{2s+2}}{Q_r^{[2s+2]}(0)Q_r^{[2s]}(0)} - \frac{(-1)^{2sL}\kappa}{Q_r^{[-2s-2]}(0)Q_r^{[-2s]}(0)} \right] \tag{64}$$
$$+ \text{(regular terms)}.$$

Thus the regularity of $\mathbf{B}(u)$ at $u = 0$ is equivalent to the condition that

$$\kappa^{2s+1} = (-1)^{2sL} \frac{Q_r^{[2s+2]}(0)Q_r^{[2s]}(0)}{Q_r^{[-2s-2]}(0)Q_r^{[-2s]}(0)}. \tag{65}$$

This is just the physical condition for singular solutions. For the twistless limit $\kappa \to 1$, this condition goes to known results given in [33].

In summary, we take a common assumption that twisted boundary conditions is a strong enough regularization scheme such that there is no repeated root if $Q_{0,1}/\gamma$ is required to be a polynomial and then derive the physical condition (65) for singular solutions.

## 3.6 Numerical results

For the rational $Q$-system, the physical condition corresponds to the polynomiality of functions $Q_{0,1}/\gamma$ and $Q_{0,2s+1}$. The numerical results show that for twisted case, *i.e.*, $\theta \neq 0$, there is no physical repeated solution and no physical singular solution, which means that for all repeated solutions and all singular solutions, either $Q_{0,1}/\gamma$ or $Q_{0,2s+1}$ is not polynomial. To be more concrete, the numerical evidence shows that for twisted boundary conditions, the polynomiality of $Q_{0,1}/\gamma$ rules out all repeated solutions, and the polynomiality of both $Q_{0,1}/\gamma$ and $Q_{0,2s+1}$ rules out all singular solutions. The relation among Bethe solutions for twisted and the periodic boundary condition is shown in the diagram 3. Examples of numerical results for Bethe solutions are given in Table 4 and Table 5.[14]

Please note that the number of primary states with respect to $SU(2)$ symmetry is equal to the number of physical Bethe solutions which do not involve roots at infinity. For spin-$s$ twisted XXX chain where $SU(2)$ symmetry is broken into $U(1)$ symmetry,[15] the number of physical Bethe solutions is given by the formula

$$\mathcal{N}_s(L, M) = c_s(L, M), \tag{66}$$

---

[14] We checked more examples with generic twist parameter $\theta$. But it costs more time when the size of the chain grows, such as for $L = 8, M = 3, s = 1, \theta = 1/10$, the $Q$-system gives 112 physical Bethe solutions and it costs about 23 seconds.

[15] Here the 'twisted' does not include the anti-periodic boundary condition, where $\theta = \pi \pmod{2\pi}$.

where the function $c_s(L; M)$ is given by (7). For spin-$s$ XXX chain with the periodic boundary condition, which is twistless, the corresponding symmetry is $SU(2)$, thus the number of physical Bethe solutions without roots at infinity is given by

$$\mathcal{N}_s(L, M) = c_s(L, M) - c_s(L, M - 1). \tag{67}$$

Such difference corresponds to the fact that some of Bethe roots go to infinity as the twist parameter $\theta$ goes to 0, where the $U(1)$ symmetry is enhanced to $SU(2)$. Physically speaking, some of the states becomes descendants when we take the twistless limit. There is a similar story in the open boundary cases, which we will discuss later.

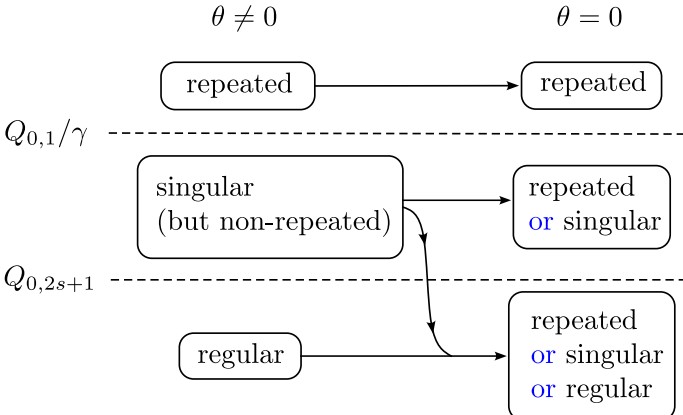

Figure 3: The relation among Bethe solutions for twisted and the periodic boundary condition. The solid line with arrow shows the change of Bethe solutions when taking the twistless limit $\theta \to 0$. The mergence of solid lines means the coincidence of solutions in the limiting process. The Bethe solutions below the dashed line satisfy the polynomiality conditions for the corresponding $Q$-functions.

Table 4: Physical Bethe solutions for $L = 4, M = 2, s = \frac{1}{2}, \kappa \equiv e^{i\theta}$. Solutions involving roots at $\infty$ correspond to the descendants with respect to the $SU(2)$ symmetry.

| $\theta = 1/100$ | $\theta = 0$ |
|---|---|
| $\{-300.00 - 173.21i, \; -300.00 + 173.21i\}$ | $\{\infty, \; \infty\}$ |
| $\{-199.50, \; -0.50001\}$ | $\{\infty, \; -0.5\}$ |
| $\{-200.00, \; 0\}$ | $\{\infty, \; 0\}$ |
| $\{-200.50, \; 0.50001\}$ | $\{\infty, \; 0.5\}$ |
| $\{0.00250 - 0.50000i, \; 0.00250 + 0.50000i\}$ | $\{-0.5i, \; 0.5i\}$ |
| $\{-0.28784, \; 0.28951\}$ | $\{-0.28868, \; 0.28868\}$ |
| 6 primary | 2 primary<br>4 descendant |

Table 5: Physical Bethe solutions for $L = 3, M = 3, s = 1$. The meaning of roots at $\infty$ is the same as Table 4.

| $\theta = 1/100$ | $\theta = 0$ |
|---|---|
| $\{-464.44, -367.78 - 350.88i, -367.78 + 350.88i\}$ | $\{\infty, \infty, \infty\}$ |
| $\{-299.71 - 173.21i, -299.71 + 173.21i, -0.57735\}$ | $\{\infty, \infty, -0.57735\}$ |
| $\{-300.29 - 173.21i, -300.29 + 173.21i, 0.57735\}$ | $\{\infty, \infty, 0.57735\}$ |
| $\{-200.86, 0.43302 - 0.55902i, 0.43302 + 0.55902i\}$ | $\{\infty, 0.43301 - 0.55902i, 0.43301 + 0.55902i\}$ |
| $\{-200.00, -0.44721i, 0.44721i\}$ | $\{\infty, -0.44721i, 0.44721i\}$ |
| $\{-199.13, -0.43302 - 0.55902i, -0.43302 + 0.55902i\}$ | $\{\infty, -0.43301 - 0.55902i, -0.43301 + 0.55902i\}$ |
| $\{0.00333 - 1.00000i, 0.00333 + 1.00000i, 0.0033333\}$ | $\{-i, i, 0\}$ |
| 7 primary | 1 primary |
| | 6 descendant |

# 4 Open boundary conditions

In this section, we will show the construction of the rational $Q$-system for spin-$s$ XXX chain with open boundaries.[16] We find that when $\alpha - \beta$, $L$ and $M$ and spin-$s$ satisfy certain relations, where $\alpha$ and $\beta$ are boundary parameters, there appears to be extra degeneracies in the spin chain, which means that there are hidden symmetries similar to that mentioned in [38]. This point is illustrated both analytically and numerically.

## 4.1 Open boundary conditions

In this case, we consider the isotropic (XXX) open Heisenberg quantum spin-$s$ chain of length $L$ with boundary magnetic fields. For example, when $s = 1$,

$$
\begin{aligned}
H = -\frac{1}{4} \sum_{n=1}^{L-1} (\mathbf{S}_n \cdot \mathbf{S}_{n+1} - (\mathbf{S}_n \cdot \mathbf{S}_{n+1})^2) \\
- \frac{1}{4}\left(\frac{1}{\alpha + \frac{1}{2}} + \frac{1}{\alpha - \frac{1}{2}}\right) S_L^z - \frac{1}{4}\left(\frac{1}{\alpha + \frac{1}{2}} - \frac{1}{\alpha - \frac{1}{2}}\right)(S_L^z)^2 \\
+ \frac{1}{4}\left(\frac{1}{\beta + \frac{1}{2}} + \frac{1}{\beta - \frac{1}{2}}\right) S_1^z - \frac{1}{4}\left(\frac{1}{\beta + \frac{1}{2}} - \frac{1}{\beta - \frac{1}{2}}\right)(S_1^z)^2,
\end{aligned}
\tag{68}
$$

where $\alpha$ and $\beta$ are boundary parameters.

The bulk $SU(2)$ symmetry is also broken to $U(1)$ symmetry on account of boundary terms in this case. Such Hamiltonian can be solved by Bethe Ansatz [30,54,55], and the corresponding Bethe equations are

$$
\frac{g(u_j - \frac{i}{2})}{f(u_j + \frac{i}{2})}\left(\frac{u_j + is}{u_j - is}\right)^{2L} = \prod_{k \neq j}^{M} \frac{(u_j - u_k + i)(u_j + u_k + i)}{(u_j - u_k - i)(u_j + u_k - i)},
\tag{69}
$$

where we have introduced functions

$$
f(u) \equiv (u - i\alpha)(u + i\beta), \qquad g(u) \equiv (u + i\alpha)(u - i\beta).
\tag{70}
$$

---

[16]The Hamiltonian of the spin-$s$ XXX chain with open boundaries can be derived by using the corresponding transfer matrix. We give the general formula for this point in appendix A.

### 4.2 The rational *Q*-system

The *QQ*-relation for the open version of the rational *Q*-system[17] is given by [27,38]

$$uQ_{1,b} = f^{[-(b-1)]}Q_{1,b-1}^+ - g^{[b-1]}Q_{1,b-1}^-,$$
$$uQ_{0,b}Q_{1,b-1} = Q_{1,b}^+Q_{0,b-1}^- - Q_{1,b}^-Q_{0,b-1}^+. \tag{71}$$

Note that the notation here has been introduced in equation (38). The boundary condition is much different from closed chains and is given by

$$Q_{0,0}(u) = \tilde{q}_\gamma(u), \qquad Q_{1,0}(u) = \tilde{Q}(u) = \prod_{j=1}^M (u-u_j)(u+u_j) = u^{2M} + \sum_{k=0}^{M-1} c_k u^{2k},$$
$$Q_{2,b}(u) = 1, \qquad b = 0,1,\ldots,M, \tag{72}$$

where we introduce functions[18]

$$\tilde{\gamma}(u) = \prod_{k=1}^{2s-1} (u-i(s-k))^{2L}, \qquad \tilde{q}_\gamma(u) = \prod_{k=0}^{2s-1} \left(u-i\left(s-k-\frac{1}{2}\right)\right)^{2L}. \tag{73}$$

Just as the twisted cases, we expect that the minimal analyticity requires $Q_{0,1}/\tilde{\gamma}$ and $Q_{0,2s+1}$ to be polynomials.

Bethe equations for open chains can be also easily derived from the *QQ*-relation for open systems. Taking $b = 1$ in (71), we have

$$uQ_{1,1} = fQ_{1,0}^+ - gQ_{1,0}^-,$$
$$uQ_{0,1}Q_{1,0} = Q_{1,1}^+Q_{0,0}^- - Q_{1,1}^-Q_{0,0}^+. \tag{74}$$

Then taking $u = u_j$ for $j = 1,\ldots,M$, we obtain the Bethe Ansatz equation,

$$\frac{g^-(u_j)Q_{0,0}^+(u_j)}{f^+(u_j)Q_{0,0}^-(u_j)} = \frac{(u_j - \frac{i}{2})}{(u_j + \frac{i}{2})}\frac{Q_{1,0}^{++}(u_j)}{Q_{1,0}^{--}(u_j)}. \tag{75}$$

### 4.3 The *P*-function and the equator

In the case of the open boundary condition, the *P*-function can be viewed as the solution for the Wronskian relation for given *Q*, which reads

$$gP^+Q^- - fP^-Q^+ = u\tilde{q}_\gamma. \tag{76}$$

Since functions *Q* and $\tilde{q}_\gamma$ are polynomials, the function *P* should also be a polynomial of *u* with the following expansion:

$$P = i\sum_{j=0}^{2sL-M} d_j u^{2j}, \tag{77}$$

where $d_j$ are coefficients determined by the Wronskian relation (76). Suppose that the *Q*-function has the expansion:

$$Q = \sum_{k=0}^M c_k u^{2k}, \tag{78}$$

---

[17]The non-physical solutions, which don't correspond to eigenstates of Hamiltonian, can be eliminated by Q-systems automatically.

[18]For simplicity, we will omit the tilde later, here we just want to emphasize that the functions $Q(u)$, $q_\gamma(u)$, $\gamma(u)$ are different from that in the twisted case.

where $c_M = 1$, then the left hand side of the Wronskian relation (76) is

$$\text{LHS} = u^{4sL+1} d_{2sL-M} c_M (-2)(2sL - 2M + \alpha - \beta) + \dots, \tag{79}$$

where ... stands for terms with order of $u$ lower than $4sL + 1$. Since the right hand side of (76) is just $\text{RHS} = u^{4sL+1} + \dots$ and we know that $c_M = 1$, we have

$$d_{2sL-M} = -\frac{1}{2(2sL - 2M + \alpha - \beta)}. \tag{80}$$

So we conclude that there will be an overall divergent factor in the function $P$ when $M = sL + \frac{1}{2}(\alpha - \beta)$. This is a special point in the sense that when magnon number $M$ is smaller than this value, there is no descendant state[19] in the Hilbert space, while there are descendant states when $M$ is larger than this value. This is what we shall call an equator and we will illustrate our statement numerically in the section 4.5.

For closed spin chains, the Wronskian relation is

$$P^+ Q^- - P^- Q^+ = q_\gamma, \tag{81}$$

where, for a given $P$, if $Q$ is a solution of (81), then $Q + \text{const} \cdot P$ also satisfies (81). However, this property does not hold for open chains whose Wronskian relation is (76), except when the magnetic parameters $\alpha$ and $\beta$ take particular values. Consider the case where $\alpha - \beta = k \in \mathbb{Z}^+$. It can be readily verified that

$$
\begin{aligned}
& g P^+ P'^- - f P^- P'^+ = 0, \\
& P' \equiv P \prod_{j=0}^{k-1} \left( u + i\left(\beta + \frac{1}{2} + j\right) \right) \left( u - i\left(\beta + \frac{1}{2} + j\right) \right).
\end{aligned} \tag{82}
$$

Consequently, for a given $P$, if $Q$ is a solution of (76) with $\alpha - \beta = k \in \mathbb{Z}^+$, then $Q + \text{const} \cdot P'$ also satisfies the equation. Moreover, if $P$ and $Q$ are polynomials, $Q + \text{const} \cdot P'$ is also a polynomial.

Assuming that $Q$ and $P$ are polynomials, the degree of $Q$ is $\deg(Q) = 2M$, while the degree of $P'$ is

$$\deg(P') = \deg(P) + 2k = 2(2sL - M) + 2(\alpha - \beta) = 2(2sL - M + \alpha - \beta). \tag{83}$$

To ensure that the degree of $Q + \text{const} \cdot P'$ remains $\deg(Q) = 2M$, a constraint arises: $\deg(P') < \deg(Q)$, which implies $M > sL + \frac{1}{2}(\alpha - \beta)$. Solutions of the form $Q + \text{const} \cdot P'$ only appear when $M$ exceeds this equator.

## 4.4 Hidden symmetries

As we mentioned before, when $\alpha - \beta$, $L$ and $M$ and spin-$s$ satisfy certain relations, there will be extra degeneracies in the spectrum of the spin chain. This suggests that there are hidden symmetries for certain parameters of the spin chain. We will illustrate this point in this part by showing that there are extra charges which commute with the transfer matrix.

We start with the following Lax operator:

$$
\mathbf{L}_{an}(u) = \begin{pmatrix} u + \frac{i}{2} + i\mathbf{S}_n^z & i\mathbf{S}_n^- \\ i\mathbf{S}_n^+ & u + \frac{i}{2} - i\mathbf{S}_n^z \end{pmatrix}_a, \tag{84}
$$

---

[19]In this open chain with finite $\alpha, \beta$, we slightly abuse the name primary states and descendant states in the sense that they refer to eigenstates organized by subspace-specific symmetry generators that commute with the Hamiltonian only within Hilbert subspaces, instead of the full Hilbert space. For more details about this point, see the section 4.4.

and the following $K$-matrices:

$$\mathbf{K}_a^-(u,\beta) = \begin{pmatrix} \mathrm{i}\left(\beta+\frac{1}{2}\right)-\left(u+\frac{i}{2}\right) & 0 \\ 0 & \mathrm{i}\left(\beta-\frac{1}{2}\right)+\left(u+\frac{i}{2}\right) \end{pmatrix}_a,$$
$$\mathbf{K}_a^+(u,\alpha) = \begin{pmatrix} \mathrm{i}\left(\alpha+\frac{1}{2}\right)+\left(u+\frac{i}{2}\right) & 0 \\ 0 & \mathrm{i}\left(\alpha-\frac{1}{2}\right)-\left(u+\frac{i}{2}\right) \end{pmatrix}_a, \tag{85}$$

where label '$a$' here means the auxiliary space. The monodromy matrix can be defined as

$$\mathbf{M}_a(u,\alpha,\beta) = \mathbf{K}_a^-(u,\beta)\prod_{n=1}^{L}\mathbf{L}_{an}(u)\mathbf{K}_a^+(u,\alpha)\prod_{n=L}^{1}\mathbf{L}_{an}(u)$$
$$= \mathbf{K}_a^-(u,\beta)\begin{pmatrix} \mathbb{A}(u,\alpha) & \mathbb{B}(u,\alpha) \\ \mathbb{C}(u,\alpha) & \mathbb{D}(u,\alpha) \end{pmatrix}_a = \mathbf{K}_a^-(u,\beta)\mathbb{M}_a(u,\alpha). \tag{86}$$

The double-row monodromy matrix $\mathbb{M}_a(u,\alpha)$ here satisfies the following relation [54]

$$\mathbf{R}_{ab}(u-v)\mathbb{M}_a(u,\alpha)\mathbf{R}_{ab}(u+v-i)\mathbb{M}_b(v,\alpha) = \mathbb{M}_b(v,\alpha)\mathbf{R}_{ab}(u+v-i)\mathbb{M}_a(u,\alpha)\mathbf{R}_{ab}(u-v), \tag{87}$$

where the $R$-matrix here is spin-1/2 representation. So one finds that the operators $\mathbb{A},\mathbb{B},\mathbb{C},\mathbb{D}$ satisfy the commutation relations:

$$\mathbb{B}(u,\alpha)\mathbb{A}(v,\alpha) - \frac{(u+v)(u-v)}{(u-v+\mathrm{i})(u+v-\mathrm{i})}\mathbb{A}(v,\alpha)\mathbb{B}(u,\alpha)$$
$$= \frac{\mathrm{i}\mathbb{B}(v,\alpha)\mathbb{A}(u,\alpha)}{u-v+\mathrm{i}} + \frac{\mathrm{i}(u-v)\mathbb{B}(v,\alpha)\mathbb{D}(u,\alpha)}{(u-v+\mathrm{i})(u+v-\mathrm{i})}, \tag{88}$$

and

$$\mathbb{D}(u,\alpha)\mathbb{B}(v,\alpha) - \frac{(u-v+\mathrm{i})(u+v-\mathrm{i})}{(u+v)(u-v)}\mathbb{B}(v,\alpha)\mathbb{D}(u,\alpha)$$
$$= \frac{\mathbb{A}(u,\alpha)\mathbb{B}(v,\alpha)}{(u+v-\mathrm{i})(u-v)} - \frac{\mathrm{i}(u+v)\mathbb{B}(u,\alpha)\mathbb{D}(v,\alpha)}{(u+v-\mathrm{i})(u-v)}$$
$$+ \frac{\mathrm{i}(u-v+\mathrm{i})^2}{(u-v)^2(u+v)}\mathbb{B}(u,\alpha)\mathbb{A}(v,\alpha) - \frac{\mathrm{i}\mathbb{B}(v,\alpha)\mathbb{A}(u,\alpha)}{(u-v)^2(u+v)}. \tag{89}$$

Then the transfer matrix is given by

$$\mathbf{T}(u,\alpha,\beta) = \mathrm{tr}_a\,\mathbf{M}_a(u,\alpha,\beta) = (-u+\mathrm{i}\beta)\,\mathbb{A}(u,\alpha) + (u+\mathrm{i}\beta)\,\mathbb{D}(u,\alpha). \tag{90}$$

By using these commutation relations, one can prove that the eigenstates of the transfer matrix (or Hamiltonian) are given by

$$|\{u_j\}_{j=1}^{M}\rangle_\alpha = \prod_{j=1}^{M}\mathbb{B}\left(u_j,\alpha\right)|\uparrow\uparrow\dots\uparrow\rangle, \tag{91}$$

where parameters $\{u_j\}_{j=1}^{M}$ satisfy the Bethe equation (69). This is called the Bethe state. The transfer matrices are in involution:

$$[\mathbf{T}(u,\alpha,\beta),\mathbf{T}(v,\alpha,\beta)] = 0, \qquad \forall u,v \in \mathbb{C}, \tag{92}$$

which makes it possible to generate conserved charges. This is a general property for integrable spin chains. However, there can be more conserved charges when certain conditions are satisfied here. To find these charges, one can define that[20]

$$\hat{\mathfrak{q}}_\alpha = \lim_{u\to\infty} \frac{\mathbb{B}(u,\alpha)}{(u+i/2)^{2L-1}}\,. \tag{93}$$

The expansion of the operators in the limit of $u \to \infty$ is given by

$$\begin{aligned}
\mathbb{A}(u,\alpha) &= (u+i/2)^{2L+1}\left(\mathbb{A}^{(1)}_\infty + \frac{1}{(u+i/2)}\mathbb{A}^{(2)}_\infty + O\left(\frac{1}{(u+i/2)^2}\right)\right),\\
\mathbb{D}(u,\alpha) &= (u+i/2)^{2L+1}\left(\mathbb{D}^{(1)}_\infty + \frac{1}{(u+i/2)}\mathbb{D}^{(2)}_\infty + O\left(\frac{1}{(u+i/2)^2}\right)\right),\\
\mathbb{B}(u,\alpha) &= (u+i/2)^{2L-1}\left(\mathbb{B}^{(1)}_\infty + O\left(\frac{1}{(u+i/2)}\right)\right),\\
\mathbb{C}(u,\alpha) &= (u+i/2)^{2L-1}\left(\mathbb{C}^{(1)}_\infty + O\left(\frac{1}{(u+i/2)}\right)\right).
\end{aligned} \tag{94}$$

According to the definition of monodromy matrix, we can derive that

$$\begin{aligned}
\mathbb{A}^{(1)}_\infty &= 1\,, & \mathbb{A}^{(2)}_\infty &= 2i\mathbf{S}^z + i(\alpha+1/2)\,,\\
\mathbb{D}^{(1)}_\infty &= -1\,, & \mathbb{D}^{(2)}_\infty &= 2i\mathbf{S}^z + i(\alpha-1/2)\,.
\end{aligned} \tag{95}$$

And the others are identical with [38]. Note that the only difference is that we keep the order of $u + i/2$ rather than $u$, and the difference of $\alpha \pm 1/2$ is cancelled in the process of the derivation of commutation relations between $\hat{\mathfrak{q}}_\alpha$ and $\mathbb{A}(v,\alpha)$, also $\mathbb{D}(v,\alpha)$. Then it can be proved that [38]

$$[\hat{\mathfrak{q}}_\alpha, \mathbb{A}(v,\alpha)] = \mathbb{B}(v,\alpha)\left((2iv+1)-2(2S^z+\alpha)\right), \tag{96}$$

$$[\hat{\mathfrak{q}}_\alpha, \mathbb{D}(v,\alpha)] = -\mathbb{B}(v,\alpha)\left((3-2iv)-2(2S^z+\alpha)\right), \tag{97}$$

$$[\hat{\mathfrak{q}}_\alpha, \mathbb{B}(v,\alpha)] = 0\,, \qquad [\hat{\mathfrak{q}}_\alpha, S^z] = \hat{\mathfrak{q}}_\alpha\,. \tag{98}$$

Here $S^z \equiv \sum_{i=1}^{L} \mathbf{S}^z_i$. Using the expression (90), we have

$$[\hat{\mathfrak{q}}_\alpha, \mathbf{T}(v,\alpha,\beta)] = \mathbb{B}(v,\alpha)4v(2S^z+\alpha-\beta-1)\,. \tag{99}$$

Acting both sides of this equation on the Bethe state $|\{u_j\}_{j=1}^{M}\rangle_\alpha$, we have

$$[\hat{\mathfrak{q}}_\alpha, \mathbf{T}(v,\alpha,\beta)]|\{u_j\}_{j=1}^{M}\rangle_\alpha = 4v(2sL-2M+\alpha-\beta-1)\mathbb{B}(v,\alpha)|\{u_j\}_{j=1}^{M}\rangle_\alpha\,. \tag{100}$$

Thus it can be derived that

$$[\hat{\mathfrak{q}}_\alpha^p, \mathbf{T}(v,\alpha,\beta)]|\{u_j\}_{j=1}^{M}\rangle_\alpha = 4pv(2sL-2M+\alpha-\beta-p)\hat{\mathfrak{q}}_\alpha^{p-1}\mathbb{B}(v,\alpha)|\{u_j\}_{j=1}^{M}\rangle_\alpha\,. \tag{101}$$

It is easy to see that, when $p = p_0 \equiv 2sL - 2M + \alpha - \beta$,

$$[\hat{\mathfrak{q}}_\alpha^{2sL-2M+\alpha-\beta}, \mathbf{T}(v,\alpha,\beta)]|\{u_j\}_{j=1}^{M}\rangle_\alpha = 0\,. \tag{102}$$

We conclude that for $p = p_0$, $\hat{\mathfrak{q}}_\alpha^{p_0}$ is a 'charge' responsible for hidden symmetries, which can be used to generate descendant states by acting on $|\{u_j\}_{j=1}^{M}\rangle_\alpha$.

---

[20]The notation introduced here is the same as that given in [38].

Table 6: Physical Bethe solutions for $L = 4, M = 2, s = \frac{1}{2}$. In the limit of free boundary condition, there are 2 primary states and 4 descendant states.

| $(\alpha, \beta) = (10, 10)$ | $(\alpha, \beta) = (10^5, 10^5)$ | $(\alpha, \beta) = (\infty, \infty)$ |
|---|---|---|
| $\{1.133\text{e}1i,\ 2.191\text{e}1i\}$ | $\{1.126\text{e}5i,\ 2.175\text{e}5i\}$ | |
| $\{1.413\text{e}1i,\ 1.197\}$ | $\{1.414\text{e}5i,\ 1.207\}$ | |
| $\{1.418\text{e}1i, 0.207\}$ | $\{1.414\text{e}5i, 0.207\}$ | |
| $\{1.417\text{e}1i, 0.499\}$ | $\{1.414\text{e}5i, 0.500\}$ | |
| $\{0.231, 0.667\}$ | $\{0.231, 0.668\}$ | $\{0.231, 0.668\}$ |
| $\{0.713 - 0.512i,$ | $\{0.716 - 0.513i,$ | $\{0.716 - 0.513i,$ |
| $0.713 + 0.512i\}$ | $0.716 + 0.513i\}$ | $0.716 + 0.513i\}$ |
| 2 primary    4 descendant | | |

We emphasize that when $M$ is smaller than $sL + \frac{1}{2}(\alpha - \beta)$, which is the position of equator as we mentioned in the section 4.3, descendant states can exist since $p_0$ is larger than 0. This is consistent with what we discussed in the section 4.3. Also, for a state with $M$ magnons, where $M \leq sL + \frac{1}{2}(\alpha - \beta)$, the corresponding descendant state has $M + p_0 = 2sL - M + \alpha - \beta$ magnons. So the magnon numbers of these two states, $M$ and $M + p_0$, are symmetric with respect to $sL + \frac{1}{2}(\alpha - \beta)$. That is why $sL + \frac{1}{2}(\alpha - \beta)$ is named equator.[21]

To be specific, the existence of descendant states requires that:

$$0 \leq M < sL + \tfrac{1}{2}(\alpha - \beta) < M + p_0 \leq 2sL, \tag{103}$$

which ensures the consistency of magnon counting and the relation between equator, primary states and descendant states.

## 4.5 Numerical results

### 4.5.1 The free boundary limit

In this part, we present two illustrative examples of Bethe solutions derived from the rational $Q$-system. We verify that for parameters $\alpha, \beta, s, L, M$ which do NOT permit a hidden symmetries, the number of solutions is coincident with the formula (66), since the apparent symmetry of the Hamiltonian is $U(1)$.

When we consider the free boundary limit $\alpha, \beta \to \infty$, the symmetry becomes $SU(2)$. In this case, the rational $Q$-system is expected to yield a number of solutions consistent with (67). This hypothesis can be numerically validated by assigning exceedingly large values to $\alpha$ and $\beta$ and observing whether the count of solutions that remain finite corresponds with equation (67).[22] The numerical results are given in Tables 6 and 7.[23]

In the case of free boundaries, where $\alpha, \beta = \infty$, we designate as **descendant states** those Bethe states associated with Bethe solutions that incorporate roots at infinity. This concept will also be used in later discussion for hidden symmetries.

---

[21]Note that for closed spin chains with $SU(2)$ symmetry, the equator is located at $M = sL$ and the magnon numbers of the primary state and the descendant state are $M$ and $2sL - M$, respectively.

[22]Due to technical limitations, the numerical program cannot directly handle boundary parameters $(\alpha, \beta) = (\infty, \infty)$. Therefore, numerical results are presented as $(\alpha, \beta)$ increases to sufficiently large finite values, allowing us to observe the behavior approaching the free boundary limit. For the column with $(\alpha, \beta) = (\infty, \infty)$, we set $f, g = 1$ in (71).

[23]The two tables below show the physical Bethe solutions. We note that the $Q$-function is defined as $Q(u) = \prod_{j=1}^{M}(u - u_j)(u + u_j)$. For brevity, only the Bethe roots $u_j$ are displayed, while their counterparts $-u_j$ are omitted and here, e in the table denotes the use of scientific notation.

Table 7: Physical Bethe solutions for $L = 3, M = 3, s = 1$. In the limit of free boundary condition, there are 1 primary state and 6 descendant states.

| $(\alpha, \beta) = (10, 10)$ | $(\alpha, \beta) = (10^5, 10^5)$ | $(\alpha, \beta) = (\infty, \infty)$ |
|---|---|---|
| $\{1.075\text{e}1i, 1.437\text{e}1i, 3.021\text{e}1i\}$ | $\{1.061\text{e}5i, 1.414\text{e}5i, 2.979\text{e}5i\}$ | |
| $\{1.699, 1.131\text{e}1i, 2.177\text{e}1i\}$ | $\{1.732, 1.126\text{e}5i, 2.175\text{e}5i\}$ | |
| $\{0.574, 1.133\text{e}1i, 2.187\text{e}1i\}$ | $\{0.577, 1.126\text{e}5i, 2.175\text{e}5i\}$ | |
| $\{0.687 + 0.479i, 0.687 - 0.479i, 1.414\text{e}1i\}$ | $\{0.692 + 0.478i, 0.692 - 0.478i, 1.414\text{e}5i\}$ | |
| $\{1.321 - 0.595i, 1.321 + 0.595i, 1.406\text{e}1i\}$ | $\{1.336 - 0.599i, 1.336 + 0.599i, 1.414\text{e}5i\}$ | |
| $\{0.277 + 0.503i, 0.277 - 0.503i, 1.417\text{e}1i\}$ | $\{0.278 + 0.503i, 0.278 - 0.503i, 1.414\text{e}5i\}$ | |
| $\{0.810 + 1.008i, 0.810 - 1.008i, 0.834\}$ | $\{0.815 + 1.009i, 0.815 - 1.009i, 0.840\}$ | $\{0.815 + 1.009i, 0.815 - 1.009i, 0.840\}$ |

1 primary    6 descendant

### 4.5.2 Hidden symmetries, equators and boundary induced strings

In this part, we present the numerical results for number of solutions given by the rational $Q$-system. In particular, we consider cases where hidden symmetries discussed in section 4.4 and ambiguities of the $Q$-functions mentioned in 4.3 can exist.

Consider the case where $L = 4, s = 1, \alpha = 1$, in the following tables, we show the possibilities where hidden symmetries can exist, *i.e.*, where $2sL - 2M + \alpha - \beta$ is an integer and satisfies the constraint (103). As we mentioned in the section 4.3, we can also calculate the equator using the equation (80) for each case and obtain the equator located at $sL + \frac{1}{2}(\alpha - \beta)$. One can see from numerical results shown in Table 8 and Table 9 that when $M$ is larger than the equator, there are descendant states in the Hilbert space.

The cases with a box here have fewer solutions than we expect for the $U(1)$ symmetry, which implies the existence of hidden symmetries. Lines here connect the solutions corresponding to primary states and their descendants after acting hidden symmetries generators $\hat{\mathfrak{q}}_\alpha^p$. One can see that after considering these descendants, the number of solutions is correct as we expected.

In the following table, we can see that the hidden symmetries can also exist, which cause some of numbers of Bethe solution to be different from expected:

However, our investigation reveals more intriguing findings. The first notable observation is that, in certain cases, an additional degree of freedom appears in the corresponding $Q$-function, as previously discussed in section 4.3. This is indicated in the table by entries marked with a number followed by '$c$', which signifies the presence of this extra freedom. This additional degree of freedom arises because, when solving the rational $Q$-system, the number of constraints is insufficient to uniquely determine the $Q$-function, leaving some parameters undetermined. Like the XXX spin chain with $SU(2)$ symmetry, the $TQ$-equation is a second difference equation and has two solutions, one is $Q$ function and another is called $P$ function. If we consider $Q$, whose degree is beyond the equator, then $Q + const.P$ for all $const$ number is the solution of the $TQ$-equation. This is the source of '$c$'. We can consider the spin flip Hamiltonian to avoid the problem.

The second observation pertains to the emergence of solutions containing boundary induced strings.[24] When $\alpha - \beta = k \in \mathbb{Z}^+$, certain Bethe solutions exhibit a structured set of roots:

$$\left\{ i\left(\alpha - \tfrac{1}{2}\right), i\left(\alpha - \tfrac{3}{2}\right), \ldots, i\left(\alpha - \tfrac{2k-1}{2}\right) \right\},$$
$$\text{and } \left\{ -i\left(\alpha - \tfrac{1}{2}\right), -i\left(\alpha - \tfrac{3}{2}\right), \ldots, -i\left(\alpha - \tfrac{2k-1}{2}\right) \right\}, \tag{104}$$

where $i\left(\alpha - \tfrac{2k-1}{2}\right) = i\left(\beta + \tfrac{1}{2}\right)$.

---

[24]We use the term 'boundary induced strings' to emphasize that the appearance of these string-like configurations is induced by the presence of boundary magnetic fields.

Table 8: Numerical results for number of solutions. Here we take $L = 4, s = 1, \alpha = 1$ with $\beta$ going from 1 to 10. Lines are used to connect solutions corresponding to primary states and their descendants.

| $\beta$ | equator | $M=0$ | $M=1$ | $M=2$ | $M=3$ | $M=4$ | $M=5$ | $M=6$ | $M=7$ | $M=8$ |
|---|---|---|---|---|---|---|---|---|---|---|
| 10 | $-0.5$ | 1 | 4 | 10 | 16 | 19 | 16 | 10 | 4 | 1 |
| 9 | 0 | 1 | 4 | 10 | 16 | 19 | 16 | 10 | 4 | 1 |
| 8 | 0.5 | 1 | 3 | 10 | 16 | 19 | 16 | 10 | 4 | 1 |
| 7 | 1 | 1 | 4 | 9 | 16 | 19 | 16 | 10 | 4 | 1 |
| 6 | 1.5 | 1 | 4 | 6 | 15 | 19 | 16 | 10 | 4 | 1 |
| 5 | 2 | 1 | 4 | 10 | 12 | 18 | 16 | 10 | 4 | 1 |
| 4 | 2.5 | 1 | 4 | 10 | 6 | 15 | 15 | 10 | 4 | 1 |
| 3 | 3 | 1 | 4 | 10 | 16 | 9 | 12 | 9 | 4 | 1 |
| 2 | 3.5 | 1 | 4 | 10 | 16 | 3 | 6 | 6 | 3 | 1 |
| 1 | 4 | 1 | 4 | 10 | 16 | 19 | 0 | 0 | 0 | 0 |
| $U(1)$ | null | 1 | 4 | 10 | 16 | 19 | 16 | 10 | 4 | 1 |
| $SU(2)$ | 4 | 1 | 3 | 6 | 6 | 3 | null | null | null | null |

Examples of boundary induced string configurations are shown in Figure 4 and Figure 5. We choose two of Bethe solutions for $k = 2$ and $k = 4$ respectively and depict them in Figure 4 and Figure 5. In those figures, red dots denote Bethe roots included in boundary induced strings and blue roots denote Bethe roots beyond the boundary induced strings.

These root configurations, termed boundary induced strings, manifest as indivisible units: they either fully appear in a solution or are entirely absent. The length of a boundary induced string is $2k$, reflecting the integer $k = \alpha - \beta$. Notably, numerical investigations show that singular strings do not exist in open chain Bethe solutions. However, for specific parameter choices, such as $\alpha = s + \frac{1}{2}$, $\beta = -s - \frac{1}{2}$, the boundary induced string's structure mimics that of a singular string, despite their distinct physical origins.

Another interesting phenomenon is the 'alignedting' of Bethe solutions. For instance, in the case of $\beta = 0, M = 8$, we identify three solutions with $Q$-functions of the form $Q + \text{const} \cdot P'$ and one solution characterized by infinite roots. The total number of solutions is the same as that for $\beta = 0, M = 1$. Specifically, the four solutions for $\beta = 0, M = 1$ bifurcate into two distinct groups: three solutions without a boundary induced string correspond to the three solutions with $Q$-functions of the form $Q + \text{const} \cdot P'$ for $\beta = 0, M = 8$, while the remaining solution for $\beta = 0, M = 1$, which includes a boundary induced string, corresponds to the solution with infinite roots for $\beta = 0, M = 8$. We will provide a detailed explanation of these phenomenon in subsequent works.

Table 9: Numerical results for number of solutions. Here we take $L = 4, s = 1, \alpha = 1$ with $\beta$ going from $-8$ to $0$. Entries marked with a number followed by '$c$' and highlighted in blue indicate cases where the $Q$-functions have certain ambiguity and the $Q$-system does not have unique solution.

| $\beta$ | equator | $M = 0$ | $M = 1$ | $M = 2$ | $M = 3$ | $M = 4$ | $M = 5$ | $M = 6$ | $M = 7$ | $M = 8$ |
|---|---|---|---|---|---|---|---|---|---|---|
| 0 | 4.5 | 1 | 4 | 10 | 16 | 19 | $3c$ | $6c$ | $6c$ | $3c$ |
| $-1$ | 5 | 1 | 4 | 10 | 16 | 19 | 16 | $9c$ | $12c$ | $9c$ |
| $-2$ | 5.5 | 1 | 4 | 10 | 16 | 19 | 16 | $6c$ | $15c$ | $15c$ |
| $-3$ | 6 | 1 | 4 | 10 | 16 | 19 | 16 | 10 | $12c$ | $18c$ |
| $-4$ | 6.5 | 1 | 4 | 10 | 16 | 19 | 16 | 10 | $6c$ | $15c$ |
| $-5$ | 7 | 1 | 4 | 10 | 16 | 19 | 16 | 10 | 4 | $9c$ |
| $-6$ | 7.5 | 1 | 4 | 10 | 16 | 19 | 16 | 10 | 4 | $3c$ |
| $-7$ | 8 | 1 | 4 | 10 | 16 | 19 | 16 | 10 | 4 | 1 |
| $-8$ | 8.5 | 1 | 4 | 10 | 16 | 19 | 16 | 10 | 4 | 1 |
| $U(1)$ | null | 1 | 4 | 10 | 16 | 19 | 16 | 10 | 4 | 1 |
| $SU(2)$ | 4 | 1 | 3 | 6 | 6 | 3 | null | null | null | null |

# 5 Conclusions

In this work, we systematically investigate the spin-$s$ $Q$-system with twisted and open boundaries.

For twisted boundary conditions, we demonstrate that the polynomiality condition for the rational $Q$-system is equivalent to the polynomiality condition of the $P$-function. Additionally, we derive the physical conditions for Bethe solutions containing singular roots. Through numerical verification, we confirm that the rational $Q$-system reproduces the expected number of Bethe solutions, thereby validating its completeness in the twisted boundary case.

For the open boundary condition, we also confirmed the completeness of the rational $Q$-system in generating all physical Bethe solutions for general boundary parameters $\alpha, \beta$, spin-$s$, the length of chains $L$, and the number of magnons $M$. Our analysis reveals novel phenomena, including hidden symmetries that emerge when $\alpha - \beta \in \mathbb{Z}$ and (103) is satisfied, leading to a correspondence between primary and descendant states across the equator $M = sL + \frac{1}{2}(\alpha - \beta)$. This finding is supported by both analytical and numerical evidence (shown in Table 8 and Table 9), and it aligns with the discussion in [38] for the special case where $\alpha - \beta = 1$. Furthermore, we identified the existence of boundary induced strings that appear as indivisible units under specific boundary parameters, offering new insights into the interplay between symmetries and boundary conditions.

There are several future directions. A key question is to fully understand the origin and conditions under which boundary induced strings appear. A more concrete analytical framework is needed to elucidate when and how these solutions arise. Another important direction is to extend this analysis of Bethe solutions to more general spin chains with open boundaries. A natural next step is to consider open spin chains with higher-rank symmetries. Also, exploring the operator spectrum in the planar limit of the $\gamma$-deformed $\mathcal{N} = 4$ super-Yang-Mills theory [41–43, 56–61] using the rational $Q$-system would be interesting.

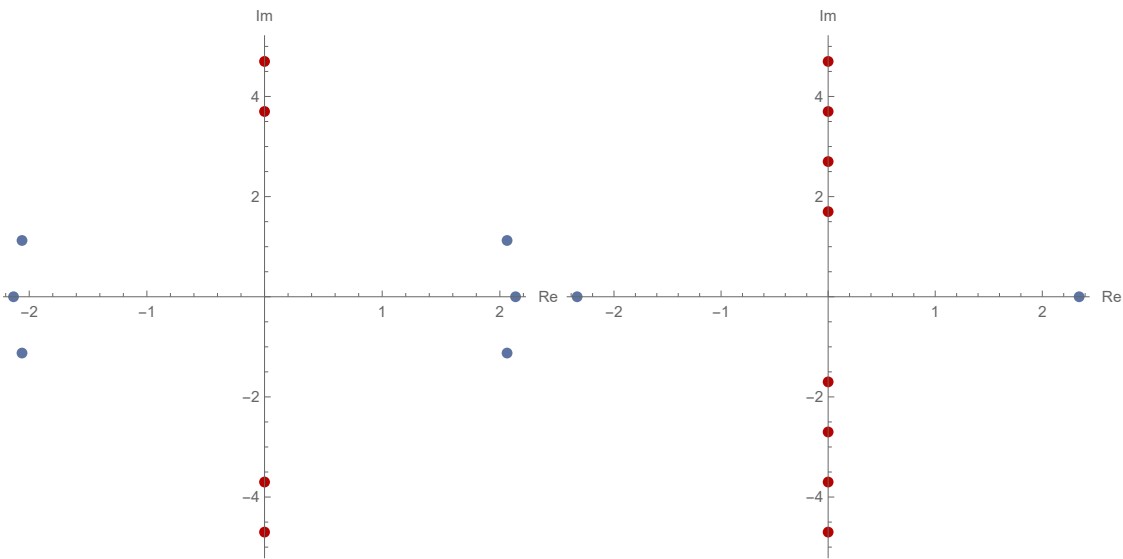

Figure 4: The string configuration of $L = 5$, $M = 5$, $s = 1$, $\alpha = 26/5$, $k = 2$.

Figure 5: The string configuration of $L = 5$, $M = 5$, $s = 1$, $\alpha = 26/5$, $k = 4$.

For the rational $Q$-system itself, there are also some interesting future works to be done. One is the generalization to spin chains with superalgebra $sl(N|M)$ in the spin-$s$ representations. Additionally, extending the polynomiality condition to more general conditions is of critical importance, as it would enable the rational $Q$-system to accommodate a broader class of integrable systems, such as the $\mathfrak{sl}(2)$ spin chain and the Lieb-Liniger model, is of critical importance. We anticipate that investigation of such systems would expand the applicability of the rational $Q$-system to a much more wider range of integrable models.

## Acknowledgments

We are extremely grateful to Yunfeng Jiang for very valuable suggestions, comments and encouragements. We also thank Junpeng Cao, Wen-Li Yang, Hongfei Shu and Yuan Miao for helpful discussions and comments.

**Funding information** Y.H. is supported by the Jiangsu Funding Program for Excellent Postdoctoral Talent.

## A  Hamiltonians for spin-$s$ XXX chains with open boundaries

In this appendix we give the general procedure to derive the Hamiltonian for spin-$s$ XXX chain with open boundaries. We start with the following propositions[25] [54]:

(a) The Lax operator $\mathbf{L}_{an}(u)$ coincides with the $R$-matrix $\mathbf{R}_{an}(u)$ in the space $W_a \otimes W_n$

$$\mathbf{L}_{an}(u) = \mathbf{R}_{an}(u). \tag{A.1}$$

(b) The value of $\mathbf{R}_{an}(u)$ at $u = 0$ is essentially the permutation operator

$$\mathbf{R}_{an}(0) = \phi(i)\mathbf{P}_{an}. \tag{A.2}$$

---

[25]The propositions and the formula of Hamiltonian holds for general spin-$s$.

(c) The value of $\mathbf{K}_1^-(u)$ at $u = 0$ is proportional to the identity

$$\mathbf{K}_1^-(0) \propto \mathbf{I}. \tag{A.3}$$

The Hamiltonian density for models with $R$-matrices satisfying conditions (a) and (b) is given by

$$h_{n,n+1} = \frac{\mathrm{d}}{\mathrm{d}u}\mathbf{R}_{n,n+1}(u)\bigg|_{u=0} \mathbf{P}_{n,n+1} = \mathbf{P}_{n,n+1}\frac{\mathrm{d}}{\mathrm{d}u}\mathbf{R}_{n,n+1}(u)\bigg|_{u=0}. \tag{A.4}$$

Then one can differentiating $\ln t(u)$ to obtain Hamiltonian

$$H \propto \frac{\mathrm{d}}{\mathrm{d}u}\ln t(u)\bigg|_{u=0}. \tag{A.5}$$

As a result, for diagonal $K$-matrices, the general Hamiltonian satisfying propositions above is given by

$$\tilde{H} = \frac{2}{\phi(i)}\sum_{n=1}^{L-1}h_{n,n+1} + \frac{2}{\phi(i)}\frac{\mathrm{tr}_a[\mathbf{K}_a^+(0)h_{a,L}]}{\mathrm{tr}_a[\mathbf{K}_a^+(0)]} + \frac{\mathrm{d}}{\mathrm{d}u}\mathbf{K}_1^-(u)\bigg|_{u=0}[\mathbf{K}_1^-(0)]^{-1}. \tag{A.6}$$

To fit in the normalized Hamiltonian, a rescaling factor is needed for specific case. The footprint $a$ and 1 here denotes the space which $\mathbf{K}_a^+(u)$ and $\mathbf{K}_1^-(u)$ act on, *i.e.*, $\mathbf{K}^+$ acts on auxiliary space and $\mathbf{K}^-$ acts on the Hilbert space of the first site. $\phi(i)$ is defined by (A.2). In particular, for the case of spin-$s$, the equation (A.2) becomes[26] [62]

$$\mathbf{R}_{an}^{(s,s)}(0) = (-1)^s(2s)!\mathbf{P}_{an}^{(s,s)}, \tag{A.7}$$

where $\mathbf{P}_{an}^{(s,s)}$ is the permutation operator in spin-$s$ representation. The spin-$s$ fused $\mathbf{R}$-matrices are given by

$$\mathbf{R}^{(s,s)} = \prod_{j=1}^{2s}(u-ji)\sum_{l=0}^{2s}\prod_{k=1}^{l}\frac{u+ki}{u-ki}\mathbf{P}^{(l)}. \tag{A.8}$$

Here $\mathbf{P}^{(l)}$ is a projector that project a direct product space of two spin-$s$ space onto a irreducible spin-$l$ space. One may also need the spin-$s$ fused $\mathbf{K}$-matrices, which are given by [55, 63–66]

$$\mathbf{K}_{\{a\}}^{-(s)}(u) = \mathbf{P}_{\{a\}}^+\prod_{k=1}^{2s}\left\{\left[\prod_{l=1}^{k-1}\mathbf{R}_{a_l a_k}^{(\frac{1}{2},\frac{1}{2})}(2u+(k+l-2s-1)i)\right]\mathbf{K}_{a_k}^{-(\frac{1}{2})}\left(u+\left(k-s-\frac{1}{2}\right)i\right)\right\}\mathbf{P}_{\{a\}}^+, \tag{A.9}$$

where the basic $R$-matrix and $K$-matrix are given by [67, 68]

$$\mathbf{R}_{a_l a_k}^{(\frac{1}{2},\frac{1}{2})}(u) = \begin{pmatrix} u+i & 0 & 0 & 0 \\ 0 & u & i & 0 \\ 0 & i & u & 0 \\ 0 & 0 & 0 & u-i \end{pmatrix}_{a_l a_k}, \qquad \mathbf{K}_{a_k}^{-(\frac{1}{2})}(u) = \begin{pmatrix} i\beta - u & 0 \\ 0 & i\beta + u \end{pmatrix}_{a_k}, \tag{A.10}$$

and the subscript $\{a\}$ denotes the direct product space of space $V_{a_l}$, $V_{a_k}$ and so on. They satisfy the boundary Yang-Baxter equation

$$\mathbf{R}_{a_1 a_2}^{(\frac{1}{2},\frac{1}{2})}(u-v)\mathbf{K}_{a_1}^{-(\frac{1}{2})}(u)\mathbf{R}_{a_1 a_2}^{(\frac{1}{2},\frac{1}{2})}(u+v)\mathbf{K}_{a_2}^{-(\frac{1}{2})}(v) = \mathbf{K}_{a_2}^{-(\frac{1}{2})}(v)\mathbf{R}_{a_1 a_2}^{(\frac{1}{2},\frac{1}{2})}(u+v)\mathbf{K}_{a_1}^{-(\frac{1}{2})}(u)\mathbf{R}_{a_1 a_2}^{(\frac{1}{2},\frac{1}{2})}(u-v). \tag{A.11}$$

Also, $\mathbf{K}_a^+(u)$ for general spin-$s$ is taken to be [54]

$$\mathbf{K}_{\{a\}}^{+(s)}(u) = \mathbf{K}_{\{a\}}^{-(s)}(-u-i)\bigg|_{\beta\to\alpha}. \tag{A.12}$$

Now we start to proof the formula (A.6).

---

[26]In the following, we will use the superscript $(s)$ for spin-$s$ case to make it clearer.

## A.1 The proof of equation (A.6)

For an integrable spin chain with open boundaries, one may construct the corresponding transfer matrix in the following way [54][27]

$$t(u) = \mathrm{tr}_a \left[ \mathbf{K}_a^+(u) \mathbf{T}_a(u) \mathbf{K}_a^-(u) \hat{\mathbf{T}}_a(u) \right], \tag{A.13}$$

where $\mathbf{T}_a(u)$ and its inversion $\hat{\mathbf{T}}_a(u)$ are defined as

$$\begin{aligned}
\mathbf{T}_a(u) &= \mathbf{R}_{a,L}(u) \mathbf{R}_{a,L-1}(u) \dots \mathbf{R}_{a,2}(u) \mathbf{R}_{a,1}(u), \\
\hat{\mathbf{T}}_a(u) &= \mathbf{R}_{a,1}(u) \mathbf{R}_{a,2}(u) \dots \mathbf{R}_{a,L-1}(u) \mathbf{R}_{a,L}(u).
\end{aligned} \tag{A.14}$$

Diffrentiating $t(u)$ at $u = 0$, we have

$$\begin{aligned}
\frac{\mathrm{d}}{\mathrm{d}u} t(u) \Big|_{u=0} =\; & \mathrm{tr}_a \left[ \frac{\mathrm{d}}{\mathrm{d}u} \big( \mathbf{K}_a^+(u) \big) \Big|_{u=0} \mathbf{T}_a(0) \mathbf{K}_a^-(0) \hat{\mathbf{T}}_a(0) \right] \\
& + \mathrm{tr}_a \left[ \mathbf{K}_a^+(0) \mathbf{T}_a(0) \frac{\mathrm{d}}{\mathrm{d}u} \big( \mathbf{K}_a^-(u) \big) \Big|_{u=0} \hat{\mathbf{T}}_a(0) \right] \\
& + \mathrm{tr}_a \left[ \mathbf{K}_a^+(0) \frac{\mathrm{d}}{\mathrm{d}u} \big( \mathbf{T}_a(u) \big) \Big|_{u=0} \mathbf{K}_a^-(0) \hat{\mathbf{T}}_a(0) \right] \\
& + \mathrm{tr}_a \left[ \mathbf{K}_a^+(0) \mathbf{T}_a(0) \mathbf{K}_a^-(0) \frac{\mathrm{d}}{\mathrm{d}u} \big( \hat{\mathbf{T}}_a(u) \big) \Big|_{u=0} \right].
\end{aligned} \tag{A.15}$$

Remembering (A.2), we can express (A.14) as

$$\begin{aligned}
\mathbf{T}_a(0) &= \phi(i)^{2L} \mathbf{P}_{a,L} \mathbf{P}_{a,L-1} \dots \mathbf{P}_{a,2} \mathbf{P}_{a,1}, \\
\hat{\mathbf{T}}_a(0) &= \phi(i)^{2L} \mathbf{P}_{a,1} \mathbf{P}_{a,2} \dots \mathbf{P}_{a,L-1} \mathbf{P}_{a,L}.
\end{aligned} \tag{A.16}$$

It is easy to deal with the first two lines of (A.15), one can just use (A.16) and move the permutation operators to the other side, then two identical permutation operators just becomes identity and contribute nothing to the trace, which is

$$\begin{aligned}
& \mathrm{tr}_a \left[ \frac{\mathrm{d}}{\mathrm{d}u} \big( \mathbf{K}_a^+(u) \big) \Big|_{u=0} \mathbf{T}_a(0) \mathbf{K}_a^-(0) \hat{\mathbf{T}}_a(0) \right] + \mathrm{tr}_a \left[ \mathbf{K}_a^+(0) \frac{\mathrm{d}}{\mathrm{d}u} \big( \mathbf{K}_a(u) \big) \Big|_{u=0} \mathbf{K}_a(0) \mathbf{K}_a^-(0) \hat{\mathbf{T}}_a(0) \right] \\
& = \phi(i)^{2L} \left( \mathrm{tr}_a \left( \frac{\mathrm{d}}{\mathrm{d}u} \big( \mathbf{K}_a^+(u) \big) \Big|_{u=0} \right) \mathbf{K}_1^-(0) + \mathrm{tr}_a \big( \mathbf{K}_a^+(0) \big) \frac{\mathrm{d}}{\mathrm{d}u} \big( \mathbf{K}_a^-(u) \big) \Big|_{u=0} \right).
\end{aligned}$$

Using (A.4), $\mathbf{T}_a'(0)$ is nothing but a summation

$$\begin{aligned}
\mathbf{T}_a'(0) &= \mathbf{R}'(0)_{a,L} \mathbf{P}_{a,L-1} \dots \mathbf{P}_{a,2} \mathbf{P}_{a,1} + \mathbf{P}_{a,L} \mathbf{R}'(0)_{a,L-1} \dots \mathbf{P}_{a,2} \mathbf{P}_{a,1} + \dots \\
&= \mathbf{P}_{a,L} h_{a,L} \mathbf{P}_{a,L-1} \dots \mathbf{P}_{a,2} \mathbf{P}_{a,1} + \mathbf{P}_{a,L} \mathbf{P}_{a,L-1} h_{a,L-1} \dots \mathbf{P}_{a,2} \mathbf{P}_{a,1} + \dots \\
&= \mathbf{P}_{a,L} \mathbf{P}_{a,L-1} \dots \mathbf{P}_{a,2} \mathbf{P}_{a,1} h_{L-1,L} + \mathbf{P}_{a,L} \mathbf{P}_{a,L-1} \dots \mathbf{P}_{a,2} \mathbf{P}_{a,1} h_{L-2,L-1} + \dots
\end{aligned} \tag{A.17}$$

Plugging this back into (A.15) and moving permutation operators to the right side, using the property (A.3) mensioned at the beginning, we obtain

$$\begin{aligned}
t'(0) =\; & \phi(i)^{2L} \left( \mathrm{tr}_a \left( \frac{\mathrm{d}}{\mathrm{d}u} \big( \mathbf{K}_a^+(u) \big) \Big|_{u=0} \right) \mathbf{K}_1^-(0) + \mathrm{tr}_a \big( \mathbf{K}_a^+(0) \big) \frac{\mathrm{d}}{\mathrm{d}u} \big( \mathbf{K}_a^-(u) \big) \Big|_{u=0} \right) \\
& + 2\phi(i)^{2L-1} \mathrm{tr}_a \left[ \mathbf{K}_a^+(0) h_{a,L} \right] \mathbf{K}_1^-(0) + 2\phi(i)^{2L-1} \mathrm{tr}_a \left[ \mathbf{K}_a^+(0) \right] \left( \sum_{n=1}^{L-1} h_{n,n+1} \right) \mathbf{K}_1^-(0).
\end{aligned} \tag{A.18}$$

---

[27] For simplicity, we omit the superscript $(s)$ of $R$-matrix and $K$-matrix in the following.

Since $t(0)$ is simply

$$t(0) = \phi(i)^{2L} \text{tr}_a[\mathbf{K}_a^+(0)]\mathbf{K}_1^-(0). \tag{A.19}$$

The final result is

$$
\begin{aligned}
\frac{\mathrm{d}}{\mathrm{d}u} \ln t(u)\bigg|_{u=0} = &\frac{2}{\phi(i)} \sum_{n=1}^{L-1} h_{n,n+1} + \frac{2}{\phi(i)} \frac{\text{tr}_a[\mathbf{K}_a^+(0)h_{a,L}]}{\text{tr}_a[\mathbf{K}_a^+(0)]} \\
&+ \frac{\mathrm{d}}{\mathrm{d}u}\mathbf{K}_1^-(u)\bigg|_{u=0} [\mathbf{K}_1^-(0)]^{-1} + \frac{\text{tr}_a[\frac{\mathrm{d}}{\mathrm{d}u}(\mathbf{K}_a^+(u))]}{\text{tr}_a[\mathbf{K}_a^+(u)]}\bigg|_{u=0},
\end{aligned}
\tag{A.20}
$$

which gives spin-$s$ Hamiltonian with diagonal open boundary.

## A.2 Hamiltonians for $s = \frac{1}{2}, 1, \frac{3}{2}$

In this part, we give the explicit form of Hamiltonian for $s = \frac{1}{2}, 1, \frac{3}{2}$ by using the general formula (A.6) [15, 16].

### A.2.1 Spin-$\frac{1}{2}$

The spin-$\frac{1}{2}$ Hamiltonian is given by

$$H = -\sum_{n=1}^{L-1} (\mathbf{S}_n \cdot \mathbf{S}_{n+1}) + \frac{1}{2\beta} S_1^z - \frac{1}{2\alpha} S_L^z + \frac{L-1}{4} \mathbf{1}_{2\times2}, \tag{A.21}$$

where we take $p_- = -i\beta$ and $p_+ = -i\alpha$ [27,62]. The expressions for the energy is given by

$$E = \sum_{j=1}^{M} \frac{2}{4u_j^2 + 1} + \frac{1}{4\beta} - \frac{1}{4\alpha}, \tag{A.22}$$

where $u_1, u_2, \ldots, u_M$ are roots for the spin-1/2 Bethe equation.

### A.2.2 Spin-1

For the case of spin-1, the diagonal fused $K$-matrix $\mathbf{K}_{\{a\}}^{-(1)}$ is given by[28] [62, 69]:

$$
\begin{aligned}
\mathbf{K}_{\{a\}}^{-(1)}(u) &= (2u+i) \begin{pmatrix} r_1(u) & 0 & 0 \\ 0 & r_2(u) & 0 \\ 0 & 0 & r_3(u) \end{pmatrix}_{\{a\}}, \\
r_1(u) &= \left( i\beta - u - \frac{i}{2} \right)\left( i\beta - u + \frac{i}{2} \right), \\
r_2(u) &= \left( i\beta - u + \frac{i}{2} \right)\left( i\beta + u - \frac{i}{2} \right), \\
r_3(u) &= \left( i\beta + u + \frac{i}{2} \right)\left( i\beta + u - \frac{i}{2} \right).
\end{aligned}
\tag{A.23}
$$

---

[28]The label $\{a\}$ here presents the direct product space of two 2-dimensional subspace.

Then the corresponding Hamiltonian is given by

$$
\begin{aligned}
H = & -\frac{1}{4} \sum_{n=1}^{L-1} (\mathbf{S}_n \cdot \mathbf{S}_{n+1} - (\mathbf{S}_n \cdot \mathbf{S}_{n+1})^2) \\
& -\frac{1}{4}\left(\frac{1}{\alpha+\frac{1}{2}} + \frac{1}{\alpha-\frac{1}{2}}\right) S_L^z - \frac{1}{4}\left(\frac{1}{\alpha+\frac{1}{2}} - \frac{1}{\alpha-\frac{1}{2}}\right)(S_L^z)^2 \\
& +\frac{1}{4}\left(\frac{1}{\beta+\frac{1}{2}} + \frac{1}{\beta-\frac{1}{2}}\right) S_1^z - \frac{1}{4}\left(\frac{1}{\beta+\frac{1}{2}} - \frac{1}{\beta-\frac{1}{2}}\right)(S_1^z)^2.
\end{aligned}
\tag{A.24}
$$

The energy in this case is given by the formula

$$
E = \sum_{j=1}^{M} \frac{1}{u_j^2 + 1} + \frac{1}{2}\left(\frac{1}{\beta-\frac{1}{2}} - \frac{1}{\alpha+\frac{1}{2}}\right),
\tag{A.25}
$$

where $u_1, u_2, \ldots, u_M$ are roots for the spin-1 Bethe equation.

### A.2.3 Spin-$\frac{3}{2}$

For the case of spin-1, the diagonal fused $K$-matrix $\mathbf{K}_{\{a'\}}^{-(\frac{3}{2})}$ is given by

$$
\begin{aligned}
\mathbf{K}_{\{a'\}}^{-(\frac{3}{2})} &= -4u(u+i)(2u+i)
\begin{pmatrix}
r_1'(u) & 0 & 0 & 0 \\
0 & r_2'(u) & 0 & 0 \\
0 & 0 & r_3'(u) & 0 \\
0 & 0 & 0 & r_4'(u)
\end{pmatrix}, \\
r_1'(u) &= (i\beta - u)(i\beta - u + i)(i\beta - u - i), \\
r_2'(u) &= (i\beta - u)(i\beta - u + i)(i\beta + u - i), \\
r_3'(u) &= (i\beta + u)(i\beta - u + i)(i\beta + u - i), \\
r_4'(u) &= (i\beta + u)(i\beta + u + i)(i\beta + u - i),
\end{aligned}
\tag{A.26}
$$

where $\{a'\}$ is just the spin-3/2 version of $\{a\}$. Note that in order to get the correct size of the boundary matrix, similarity transformation is needed in the fusion procedure. After the transformation, there will be some null rows and columns. Once removing those zeros, the size of the matrix is correct. For example, according to (A.9), the original fused $K$-matrices of spin-1 case $\mathbf{K}_{\{a\}}^{-(1)}(u)$ is a $4 \times 4$ matrix, which is not a correct size of spin-1 representation. After diagonalization, there will be a null column and row with all elements being zero, then one can remove this column to get the correct size. To be more specific, there is always a similarity transformation [62]

$$
A^{(s)} \mathbf{K}_{\{a\}}^{-(s)}(u) \left(A^{(s)}\right)^{-1},
\tag{A.27}
$$

to obtain a diagonal $K_{\{a\}}^{-(s)}(u)$, and the removal of null columns or rows to obtain a correct size. One of the similarity transformation matrices for spin-1 and spin-3/2 are respectively

$$
A^{(1)} =
\begin{pmatrix}
1 & 0 & 0 & 0 \\
0 & 1 & 1 & 0 \\
0 & -1 & 1 & 0 \\
0 & 0 & 0 & 1
\end{pmatrix},
\tag{A.28}
$$

and

$$A^{(\frac{3}{2})} = \begin{pmatrix} 1 & 0 & 0 & 0 & 0 & 0 & 0 & 0 \\ 0 & 1 & 1 & 0 & 1 & 0 & 0 & 0 \\ 0 & -1 & 1 & 0 & 0 & 0 & 0 & 0 \\ 0 & -1 & 0 & 0 & 1 & 0 & 0 & 0 \\ 0 & 0 & 0 & -1 & 0 & 1 & 0 & 0 \\ 0 & 0 & 0 & -1 & 0 & 0 & 1 & 0 \\ 0 & 0 & 0 & 1 & 0 & 1 & 1 & 0 \\ 0 & 0 & 0 & 0 & 0 & 0 & 0 & 1 \end{pmatrix}. \tag{A.29}$$

Then the resulting Hamiltonian is

$$\begin{aligned} H = & \frac{1}{432} \sum_{n=1}^{L} \left( 27 \mathbf{S}_n \cdot \mathbf{S}_{n+1} - 8(\mathbf{S}_n \cdot \mathbf{S}_{n+1})^2 - 16(\mathbf{S}_n \cdot \mathbf{S}_{n+1})^3 \right) + \frac{3L}{8} \mathbf{1}_{3\times 3} \\ & - \left( \frac{13}{24\alpha} - \frac{1}{48(\alpha+1)} - \frac{1}{48(\alpha-1)} \right) S_L^z + \left( \frac{1}{8(\alpha-1)} - \frac{1}{8(\alpha+1)} \right) (S_L^z)^2 \\ & + \left( \frac{1}{6\alpha} - \frac{1}{12(\alpha+1)} - \frac{1}{12(\alpha-1)} \right) (S_L^z)^3 + \left( \frac{13}{24\beta} - \frac{1}{48(\beta-1)} - \frac{1}{48(\beta+1)} \right) S_1^z \\ & + \left( \frac{1}{8(\beta-1)} - \frac{1}{8(\beta+1)} \right) (S_1^z)^2 - \left( \frac{1}{6\beta} - \frac{1}{12(\beta-1)} - \frac{1}{12(\beta+1)} \right) (S_1^z)^3. \end{aligned} \tag{A.30}$$

The corresponding energy is given by

$$E = \sum_{j=1}^{M} \frac{6}{4u_j^2 + 9} - \frac{1}{4\alpha} - \frac{17}{32(\alpha+1)} + \frac{1}{32(\alpha-1)} + \frac{17}{32(\beta-1)} + \frac{1}{4\beta} - \frac{1}{32(\beta+1)} + \frac{3}{8}. \tag{A.31}$$

where $u_1, u_2, \dots, u_M$ are roots for the spin-3/2 Bethe equation. For general spin-$s$ case, we conclude that the energy formula should be

$$E = \sum_{j=1}^{M} \frac{s}{u_j^2 + s^2} + E_0(\alpha, \beta), \tag{A.32}$$

where $u_1, u_2, \dots, u_M$ are roots for the spin-$s$ Bethe equation. The energy shift $E_0(\alpha, \beta)$ is independent of Bethe roots $u_j$ and can be determined by acting Hamiltonian on the ground state.

## A.3 Numerical results for energies

In this part, we present the numerical results for energies in the spin-$1/2, 1, 3/2$ cases.

One can check that the number of solutions here is consistent with the formula (66),

$$\mathcal{N}_s(L, M) = c_s(L, M). \tag{A.33}$$

Table 10: The Energy of spin-$\frac{1}{2}$ open chain with $\alpha = \frac{1}{2}$, $\beta = \frac{1}{2}$.

| L | M | Spectrum | Numbers |
|---|---|----------|---------|
| 2 | 1 | -0.61803, 1.6180 | 2 |
| 3 | 1 | -0.66170, 0.82104, 1.8407 | 3 |
| 4 | 2 | -0.76284, 0.12575, 1.0000, 1.1593, 1.5202, 2.9576 | 6 |
| 5 | 3 | -0.78836, -0.18253, 0.52889, 0.61356, 1.1148, 1.2654, 1.4037, 2.1135, 2.5659, 3.3651 | 10 |

Table 11: The Energy of spin-1 open chain with $\alpha = 1$, $\beta = 1$.

| L | M | Spectrum | Numbers |
|---|---|---|---|
| 2 | 1 | -0.79533, 0.62867 | 2 |
| 2 | 2 | -0.91844, 0.018053, 2.2337 | 3 |
| 4 | 2 | -0.98860, -0.54138, -0.15149, 0.11504, 0.40282, 0.84150, 1.1947, 1.4289, 2.1028, 2.5957 | 10 |
| 5 | 3 | -1.0755, -0.79302, -0.49073, -0.44834, -0.21830, -0.11396, -0.045126, 0.19786, 0.29370, 0.39515, 0.39723, 0.68085, 0.77866, 0.78066, 0.99690, 1.0636, 1.2974, 1.3148, 1.3164, 1.5179, 1.7087, 1.7496, 1.8494, 1.8839, 2.0654, 2.3519, 2.5897, 2.9202, 2.9872, 3.3811 | 30 |

Table 12: The Energy of spin-$\frac{3}{2}$ open chain with $\alpha = \frac{3}{5}$, $\beta = \frac{3}{5}$.

| L | M | Spectrum | Numbers |
|---|---|---|---|
| 2 | 1 | 0.45042, 1.4454 | 2 |
| 2 | 2 | 0.75984, 1.4033, 2.1285 | 3 |
| 4 | 2 | 0.24587, 0.53101, 0.74787, 0.78676, 0.87396, 1.2355, 1.6049, 1.7081, 1.9207, 2.0952 | 10 |
| 5 | 3 | 0.23891, 0.42931, 0.54419, 0.60837, 0.64456, 0.67539, 0.84227, 0.87105, 0.89532, 0.92225, 1.0966, 1.1097, 1.2040, 1.2205, 1.3160, 1.4020, 1.4054, 1.4897, 1.5845, 1.6460, 1.7757, 1.8495, 1.8807, 2.0437, 2.1231, 2.1527, 2.1656, 2.2354, 2.4532, 2.5299, 2.5390, 2.7584, 2.9172, 3.1378, 3.1672 | 35 |

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
