# Peer review of "Spin-$s$ $Q$-systems: Twist and Open Boundaries"

_SciPost Physics Core, doi:SciPost Phys. Core 8, 057 (2025)_

## Round 1 · Referee Report · Anonymous (Referee 2) · 2025-6-22

Strengths

  1. It extends the Q-system method to twisted boundary conditions in spin s XXX chain and determines the physicality requirement for Bethe roots.
  2. In boundary spin s XXX chains with specific diagonal boundary conditions it identifies a new type of string solution and hidden symmetries

Weaknesses

  1. As I emphasized in my report it would have improved the paper considerably if proofs of the rationality of the Q-system vs. physicality of the Bethe roots could have been derived.

Report

The manuscript meets all the SciPost Physics Core's expectations and general acceptance criteria.

I understand that the authors do not want to extend the manuscript with proofs concerning the relation between Q-systems and physical solutions. Although I believe it would have been improved the paper considerably.

I am satisfied with the clarification they introduced at various parts and I am supporting the publication in SciPost Physics Core after a few minor corrections.

Requested changes

In the newly introduced part some inconsistencies appeared:

  1. in (2.1) the sum should go up to $L$, otherwise it would be the boundary Hamiltonian

  2. below (3.2) the authors write: "with $S_{\pm n} = S^x_n \pm iS^y_ n$ and the periodic boundary condition". Do they mean by the periodic boundary condition $S_n=S_{n+L}$? I think it should be written out.

  3. Below (3.3) "For $L = 5, M = 2, s = 1/2, \phi= 1/3$, there are 16 solutions from Bethe ansatz equations. " What is $\phi$ here? Is it $\theta$? See also table 3.

  4. In many (important) places the authors write "Numerical evidence shows that..." e.g. concerning polynomiality of $T_0/\gamma$ and absence of repeated roots. It would be nice to spell out a bit in which extent they investigated numerically the problem. For small system sizes? For specific or generic twists?

  5. I also found one more imprecise statement in the introduction:

"Notably, in the planar limit of N = 4 super-Yang-Mills theory, the one-loop anomalous dimensions of single-trace operators map directly to the spectrum of this spin chain [2]." This is not true for all operators, but only for some of them.

Recommendation

Publish (meets expectations and criteria for this Journal)

  • validity: high
  • significance: good
  • originality: high
  • clarity: good
  • formatting: good
  • grammar: reasonable

Author:  Jue Hou  on 2025-07-05  [id 5619]

(in reply to Report 1 on 2025-06-22)
Category:
answer to question

We would like to express our gratitude to the Referee #2 for the report and valuable suggestions. We have revised our manuscript as follows: 1. All typos identified in Requested changes 1 and 3 have been corrected and the explicit form of the periodic boundary condition in Requested changes 2 has been written out. Thank the referee for pointing them out. 2. In response to your inquiry about our numerical evidence: Regarding the absence of repeated roots, we have added Footnote 10 to provide a detailed explanation of our numerical investigations. For the numerical results concerning Bethe solutions, we have included Footnote 14 in Section 3.6, which offers more details about our findings. 3. We revised the statement about the anomalous dimensions to ensure it exact.

---

## Round 1 · Referee Report · Anonymous (Referee 1) · 2025-7-2

Strengths

1- The article studies the $Q$-system for spin-$s$ XXX model with twists and open boundaries systematically.

2- The article discussed the hidden symmetries in the open boundary cases.

Weaknesses

1- Some parts are not explained in a elucidative way. For example, true physical solutions to the "ambiguous Q-functions" need to be investigated and explained.

2- The term ``magnetic string'' is still inappropriate in my opinion. I give the suggestion in the report part.

Report

Things that need to be improved:

1- Page 6, Sec. 2.2. The authors used "$\mathcal{T}$Q-relation''. This is not appropriate in my opinion. Because Eq. (2.6) is the usual TQ relation in the literature. The authors used "$\mathcal{T}$" to avoid my previous comment that the so-called T-series defined in Sec. 2.4 (page 9). This is probably fine. But Eq. (2.6) is different from Eq. (2.18) with $\mathcal{T}_0$. The polynomials $\tau$ and $\mathcal{T}_0$ only coincide when $s=1/2$, and they differ by a polynomial $\gamma$ when $s \geq 1$. I would suggest that the relation that $\tau$ satisfies should be addressed as "TQ-relation", and the ones that $\mathcal{T}_n$ satisfy should be addressed as "$\mathcal{T}$Q-relations". I suggest that the authors to check this thoroughly in the draft.

2- Page 10, below Eq. (3.1), the operators $S^{\pm, z}_n$ are generators of Lie algebra $\mathfrak{sl}_2$ instead of $\mathfrak{su}_2$.

3- I'm still not convinced by the authors about the term "magnetic strings". As the authors explained, the point is that those Bethe roots are associated with the boundary magnetic fields. The emphasis should be on $\mathit{boundary}$ instead of $\mathit{magnetic}$. I would suggest the authors to replace the term "magnetic strings" by "$\textit{boundary induced strings}$", which makes more sense in my opinion.

4- Page 25, Sec. 4.5.1. The authors tried to discuss the free boundary limit. Actually, the whole calculation can be done exactly (instead of approximately as in the draft) by taking the limit $\alpha, \beta \to \infty$ while rescaling the transfer matrix properly. This means that the authors just need to solve the Q-system with two boundary polynomials $f,g \to 1$, which is much simpler. And all the physical solutions should be the highest-weight states of the global SU(2) symmetry, as it should be. I suggest that the authors just add the calculation with $f=g=1$, which is more conclusive in my perspective.

5- The "descendant states" in Table 8 are essentially the "primary states" with the correct number of Bethe roots at $\infty$, if I understand correctly. It seems that I cannot find such a sentence in the draft. I would suggest the authors to add such an explanation at the appropriate place.

Typos:

1- Page 11, the caption of Table 3, it should read "$\theta = \frac{1}{3}$" instead of $\phi = \frac{1}{3}$.

2- Page 21, below Eq. (4.6), it should read $Q_{0,1} / \tilde{\gamma}$, instead of $Q_{0,1} / \gamma$.

3- Page 22, below Eq. (4.13), it should read "we shall call" instead of "we shell call".

Requested changes

See the report part.

Recommendation

Ask for minor revision

  • validity: high
  • significance: good
  • originality: high
  • clarity: ok
  • formatting: good
  • grammar: good

Author:  Jue Hou  on 2025-07-05  [id 5620]

(in reply to Report 2 on 2025-07-02)
Category:
answer to question

We would like to express our gratitude to the Referee #1 for the report and valuable suggestions. We have revised our manuscript as follows: 1. We check the notations thoroughly and correct them. We also add Footnotes 2 and 7 to explain the difference. 2. The algebra is corrected. 3. We have changed the term 'magnetic string' into 'boundary induced string'. 4. Page 25, Sec. 4.5.1, we update the data with $\alpha,\beta=\infty$ and update the Footnote 22 as an explanation. 5. We add some explanations for states mentioned in Table 8. We correct all the typos mentioned in the report.

---

## Round 2 · Referee Report · Anonymous (Referee 1) · 2025-7-19

Strengths

See previous report.

Weaknesses

See previous report.

Report

I'm satisfied with the current version, which implemented my suggestions adequately. I suggest the draft to be published in SciPost Physics Core in the current form.

Recommendation

Publish (meets expectations and criteria for this Journal)

---

## Editorial Decision

published